# Geostationary Environment Monitoring Spectrometer (GEMS) polarization characteristics and correction algorithm

Haklim Choi [1,a], Xiong Liu [2], Ukkyo Jeong [3,*], Heesung Chong [4,b], Jhoon Kim [4], Myung Hwan Ahn [5], Dai Ho Ko [6], Dong-won Lee [7], Kyung-Jung Moon [7], and Kwang-Mog Lee [1,*]

[1]Department of Astronomy and Atmospheric Sciences, Kyungpook National University, Daegu, Republic of Korea
[2]Center for Astrophysics | Harvard & Smithsonian, Cambridge, MA, USA
[3]Division of Earth Environmental System Science, Major of Spatial Information Engineering, Pukyong National University, Busan, Republic of Korea
[4]Department of Atmospheric Sciences, Yonsei University, Seoul, Republic of Korea
[5]Department of Climate and Energy Systems Engineering, Ewha Womans University, Seoul, Korea`
[6]Korea Aerospace Research Institute, Daejeon, Korea
[7]National Institute of Environmental Research, Environmental Satellite Center, Incheon, Republic of Korea
[a]now at: Kyungpook Institute of Oceanography, Kyungpook National University
[b]now at: Center for Astrophysics | Harvard & Smithsonian

*Correspondence to*: Kwang-Mog Lee (kmlee@knu.ac.kr) and Ukkyo Jeong (ukkyo.jeong@pknu.ac.kr)

**Abstract.**

The Geostationary Environment Monitoring Spectrometer (GEMS) is the first geostationary earth orbit (GEO) environmental instrument, onboard the Geostationary Korea Multi-Purpose Satellite–2B (GEO-KOMPSAT-2B) launched on 19 February 2020, and is measuring reflected radiance from the Earth's surface and atmosphere system in the range of 300 to 500 nm in the ultraviolet-visible (UV-Vis) region. The radiometric response of a satellite sensor that measures the UV-Vis wavelength region can depend on the polarization states of the incoming light. To reduce the sensitivity due to polarization, many current low earth orbit (LEO) satellites are equipped with a scrambler to depolarize the signals or a polarization measurement device (PMD) that simultaneously measures the polarization state of the atmosphere, then utilizes it for a polarization correction. However, a novel polarization correction algorithm is required since GEMS does not have a scrambler or a PMD. Therefore, this study aims to improve the radiometric accuracy of GEMS by developing a polarization correction algorithm optimized for GEMS that simultaneously considers the atmosphere's polarization state and the instrument's polarization sensitivity characteristics. The polarization factor and axis were derived by the preflight test on the ground as a function of wavelengths, showing a polarization sensitivity of more than 2% at some specific wavelengths. The polarization states of the atmosphere are configured as a look-up table (LUT) using the Vector Linearized Discrete Ordinate Radiative-Transfer model (VLIDORT). Depending on the observation geometry and atmospheric conditions, the observed radiance spectrum can include a polarization error of 2%. The performance of the proposed GEMS polarization algorithm was assessed using synthetic data, and the errors due to polarization were found to be larger in clear regions than in cloudy regions. After the polarization correction, polarization errors were reduced close to zero for almost all wavelengths, including the wavelength regions with high peaks and curvatures

in the GEMS polarization factor, which sufficiently demonstrates the effectiveness of the proposed polarization correction algorithm. From the actual observation data after the launch of GEMS, the diurnal variation for the spatial distribution of polarization error was confirmed to be minimum at noon and maximum at sunrise/sunset. This can be used to improve the quality of GEMS measurements, the first geostationary environmental satellite, and then contribute to the retrieved accuracy

of various Level 2 products (hereafter, L2), such as trace gases and aerosols in the atmosphere.

## 1 Introduction

The ultraviolet−visible (UV−Vis) light in the Earth's atmosphere originates from the sun, and the radiation energy emerging from the atmosphere and surface and measured in space depends on the number of photons that are scattered by air molecules, aerosols, and clouds and absorbed by trace gases such as ozone ($O_3$), sulfur dioxide ($SO_2$), nitrogen dioxide ($NO_2$),

formaldehyde (HCHO), and chlorine dioxide (OClO) or reflected from the surface. Upon reaching the Earth-atmosphere system, the unpolarized sunlight becomes partially polarized as it interacts with the atmosphere. Many previous studies evaluated the effect of polarization on radiance intensity. The error caused by neglecting the polarization can reach up to 10% within UV-Vis regions (Mishchenko et al., 1994; Lacis et al., 1998; Kotchenova et al., 2006). Thus, the polarization of light must be taken into account for the retrieval of aerosol compositions and trace gases in the atmosphere (Natraj et al., 2008;

Stam et al., 1999). Additionally, understanding the influence of polarization caused by atmospheric compositions in calculating satellite signals is a significant challenge (Dubovik et al., 2019).

Many satellite sensors onboard low earth orbit (LEO), such as the Ozone Monitoring Instrument (OMI; Levelt et al., 2018), Global Ozone Monitoring Experiment (GOME; Burrow et al., 1999), GOME-2 (Callies et al., 2000; Munro et al., 2016), the Scanning Imaging Absorption Spectrometer for Atmospheric Chartography (SCIAMACHY; Bovensmann et al., 1999), Ozone

Mapping and Profiler Suite (OMPS; Flynn et al., 2006 ), and TROPOspheric Monitoring Instrument (TROPOMI; Veefkind et al., 2012) measure the solar radiance in the UV-Vis spectral range. The Geostationary Environment Monitoring Spectrometer (GEMS; Kim et al., 2020) was launched on 19 February 2020 onboard the Geostationary Korea Multi-Purpose Satellite–2B (GEO-KOMPSAT-2B) and measure the reflected radiance from the Earth-atmosphere system in the UV-Vis region from 300 to 500 nm and 0.2 nm sampling with a resolution of 0.6 nm (Kim et al., 2020). Further, GEMS in South Korea and Tropospheric

Emissions: Monitoring of Pollution (TEMPO; Zoogman et al., 2017) in the United States, and Sentinel-4 (Ingmann et al., 2012) in Europe, jointly comprise a Geostationary-Air Quality (Geo-AQ) constellation to monitor the long- and short-range transport of pollutants across the Atlantic and Pacific Oceans in the Northern Hemisphere.

To retrieve the pollutant products from the satellite, accurate and stable measurement of reflected radiance is imperative. There are various sources of errors in the measured radiance spectrum. One of these is the polarization of light reaching the instrument

onboard the spacecraft since polarization affects the magnitude of the measured radiance. The radiometric response of a satellite instrument depends on the polarization state of the incoming light caused by mirrors, gratings, and prisms (Schutgens and Stammes, 2003). There are some approaches to reducing the polarization sensitivity of an instrument: the first is a

depolarization method that destroys the polarization information by scrambling, as is done for the TROPOspheric Monitoring Instrument (TROPOMI), OMI, the Solar Backscatter UltraViolet instrument and Total Ozone Mapping Spectrometer (SBUV/TOMS; Heath et al., 1975). The second is a polarization characterization method that measures the instrument polarization sensitivity and atmospheric polarization and is used for GOME, GOME-2, and SCIAMACHY. These satellites primarily measure the polarization state to improve radiometric accuracy (Bovensmann et al., 1999; Burrows et al., 1999; Callies et al., 2000). GOME, GOME-2, and SCIAMACHY, which are all equipped with the Polarization Measurement Device (PMD), correct the polarization error using Stokes fraction ($Q/I$) measured by PMD (Krijger et al., 2005; Liebing et al., 2018). The polarization response is determined on-ground using the PMD. The single-scattering parameterization method is used at wavelengths that are not observed by the PMD (Stammes et al., 1997; Schutgens and Stammes, 2002 and 2003). Stam et al. (2000) mentioned that, for polarization-sensitive instruments, the best way to minimize errors in the observed radiance is by measuring the state of polarization of the incident light simultaneously with the radiances. However, unfortunately, GEMS does not have a sensor that detects the polarization states of the atmosphere, and the scrambler is difficult to implement in a large-aperture instrument such as GEMS. So, it is impossible to utilize the two representative methods. Besides, the optical sensor can be designed to be relatively insensitive to the polarization state of the incoming radiation by including a polarization compensator in the optical train to offset the polarization sensitivity caused by the remaining optical train in the sensor. But,this approach is not practical or effective for the GEMS platform.

Therefore, since GEMS requires an optimized polarization correction algorithm using a separate approach other than these two methods, we developed a polarization correction algorithm based on the simulation results from the radiative transfer model (RTM) and the polarization sensitivity of instrument. In terms of a similar approach, the Moderate Resolution Imaging Spectroradiometer (MODIS) and Visible Infrared Imaging Radiometer Suite (VIIRS) instruments also lack both scramblers and PMDs. The polarization characteristics are measured during pre-haunch polarization testing on the ground. (Gordon et al., 1997; Meister et al., 2005; Sun et al., 2016). For example, MODIS, whose polarization sensitivity is up to 5.4% for certain bands, can produce radiance errors of up to 2.7% (Meister et al., 2005 and 2006). For these instruments, on-orbit polarization correction using a pre-constructed polarization coefficient database based on the Mueller matrix is derived from linear Stokes vector components modeled from a Second Simulation of the Satellite Signal in the Solar Spectrum-Vector version (6SV; Kotchenova et al., 2006 and 2007). This is a basic vector version of the RTM for the calculation of a look-up table (LUT) in the MODIS atmospheric correction algorithm. Likewise, in this study, we accomplish a polarization sensitivity analysis based on a GEMS polarization test. To determine the degree of linear polarization (DoLP) of the light that is incident to the instrument, the Stokes parameters ($Q$, and $U$) for various atmospheric conditions were included in the LUT. The polarization state of the back-scattered sunlight that enters the GEMS sensor from the Earth-atmosphere system was calculated by constructing a radiative transfer model based on the Vector Linearized Discrete Ordinate Radiative Transfer model (VLIDORT; Spurr et al., 2006; Spurr and Christi, 2019) which could simulate the spectral range of GEMS. VLIDORT has been benchmarked and verified through various models within UV-Vis region (Escribano et al., 2019; Korkin et al., 2020).

In this study, we describe the polarization correction algorithm for GEMS. The GEMS polarization correction algorithm has been developed alongside a cloud top pressure retrieval algorithm to account for the cloud region. This is unlike the polarization correction algorithm of other satellites based on the LUT method with considering the Rayleigh atmosphere under clear-sky conditions. In the following sections, in Section 2, we introduce the GEMS polarization characteristics determined by the on-ground polarization test pre-launch. In Section 3, the methodology and auxiliary data used for the polarization correction algorithm of GEMS are described, and Section 4 shows the evaluated results applied to synthetic data and actual observation data.

## 2 Instrumentation

### 2.1 Overview of GEMS

GEMS, a geostationary environmental satellite instrument, is a UV-Vis hyper-spectrometer sensor mounted on the GeoKOMPSAT-2B. The GEMS instrument was co-developed by the Korea Aerospace Research Institute (KARI) and the Ball Aerospace and Technologies Corporation (BATC) in Boulder, Colorado. Details of the GEMS mission, including the spacecraft, scientific products, and applications, are described in Kim et al. (2020). Thus, we briefly introduce the overview of GEMS here. The spacecraft is located about 36,000 km above the equator at 128.2°E and is primarily intended for atmospheric observation in Asia. The field of regard of GEMS is from 5°S to 45°N and extends from the longitude of India (75°E) to the west to that of Japan (145°E) to the east. The spectral range of GEMS is 300 to 500 nm, and observations are only made during the day at 1-hour intervals (about 8 times per day). The typical products of GEMS include aerosol properties, $O_3$, $NO_2$, $SO_2$, HCHO, cloud information, and the UV index. The specifications of the GEMS instrument and its characteristics are briefly summarized in Table.1.

### 2.2 Polarization Characteristics of GEMS

To obtain accurate radiometric data at the top of the atmosphere (TOA), it is essential to understand the polarization sensitivity information. The incident light to the GEMS payload reaches the spectrometer passing through the telescope optics that consist of a scan mirror, Schmidt mirror, and projection mirror. The Offner-type spectrometer consists of a slit, a waveplate, a grating, and other components. Then, the diffracted light is projected to the Charge-Coupled Device (CCD) at the Focal Plane Assembly (FPA). It is challenging to observe the characteristics of the sensor when operating in orbit. Thus, a polarization test of GEMS to identify the intrinsic polarization sensitivities was performed on the ground before the launch by BATC as in other studies for various instruments (Sun and Xiong, 2007; McIntire et al., 2016; Liebing et al., 2018). The configuration of the polarization test is depicted in Figure 1. The GEMS instrument is located inside the thermal vacuum chamber (TVAC), and a wire-grid VersalightTM polarizer (Baur, 2003) is placed in the illumination path between the large spherical source (LSS) Integrating Sphere and the GEMS instrument. The polarizer sheet was rotated from 0° to 720° in 5° increments. Here, zero degrees corresponds to the negative gravity vector. The polarization test was repeated 10 times for the same polarizer angle at the fixed

scan mirror assembly (SMA) position. The fixed SMA angle is 0˚, which represents the nominal position. Note that a deviation of the SMA from 0˚ position induces a shift in the entire view toward the north or south, thereby diverging from nominal operations. The polarization test images were collected for 60 co-added frames for each polarizer angle position. The setup environment for the polarization test of GEMS is summarized in Table 2. The polarization factor (PF), also known as the linear polarization sensitivity (LPS), and the polarization axis (PA) as a function of wavelengths can be derived for each cross-track position using the Fourier transform method (Moyer et al., 2017) on signals obtained from the polarization test. PF represents the sensitivity of an optical system to polarization, expressed as a percentile, while PA indicates the axis at which the maximum transmission occurs. However, the signals are reduced as the distance get increase toward the edges (which represent the North/South direction) from the central position of the CCD (here in fixed as 0˚), which is the angle at which the SMA is located. This implies that the response sensitivity to the polarization source from the integrating sphere decreases not only in the North/South direction but also across the wavelength spectrum on the CCD, making it difficult to reliably detect a consistent signal. Accordingly, the quality of the estimated result can be deteriorated.

Figure 2 shows the derived PF and PA from the polarization test, respectively. Over the majority of wavelength ranges, the PF stays within the GEMS expected range of 2%. Nevertheless, several significant PF features, such as three bump points (2.59 % at 432 nm, 2.23% at 454.6 nm, and 3.46% at 494.8 nm, respectively) and sharp inflection around 350 nm of the PF spectrum occur at specific wavelengths. The increment of PF at a certain wavelength is associated with the coating material of the Schmidt mirror of the telescope. The Schmidt mirror was multi-layer coated to attenuate the effects of stray light and involves a risk of a change in the transmittance. The GEMS PF increases at the wavelength where the transmittance of the Schmidt mirror coating decreases. This is an inevitable result. Another point is the stray light feature (Zong et al., 2007) that appears at short wavelengths below 350 nm which presented like a jagged curve. This fluctuated characteristic corresponds to the long-wavelength stray light incident on the short-wavelength section of the GEMS FPA. The effects of stray light and Schmidt mirror coating have generated uneven and curved PA and PF spectra. As a result, the observed radiance spectrum response is non-uniform across wavelength due to the non-uniform polarization characteristics (PF and PA), which can lead to degraded performance of the retrieval algorithms.

## 3 Methodology and Auxiliary Data

### 3.1 Polarization Correction Equation

As mentioned in the previous section, the UV-Vis spectrometer is affected by the polarization state of incoming light. According to Sun and Xiong (2007), the intensity of the signal that reaches the detector is defined as follows if the incident light is linearly polarized:

$$I_{polcor} = \frac{I_{obs}}{1+facos[2(\chi-\phi)]} \tag{1}$$

where $I_{obs}$ is the measured radiance (L1B), $I_{polcor}$ is the true value of the radiance that corresponds to the polarization-corrected radiance. $f$ and $\phi$ are the PF and PA as determined from the on-ground polarization test of GEMS, respectively. $a$ is the DoLP of the atmosphere, and $\chi$ is the polarization angle relative to the Instrument Reference Plane (IRP).

The polarization angle ($\chi_{LMP}$) can be calculated by Eq. (2) using the $U$ and $Q$ components determined from the RTM as follows:

$$\chi_{LMP} = \frac{1}{2}\arctan\left(\frac{U}{Q}\right) \tag{2}$$

The polarization angle is defined relative to the local meridian plane (LMP) and ranges from −90° to 90° (Figure S1). Since the instrument reference plane (IRP) of the instrument is different from the LMP, the polarization angle at the satellite instrument is not equal to $\chi_{LMP}$. The coordinate axis was transformed to determine the IRP at the position of the polarizer sheet that is consistent with the LMP. Each component inside the satellite that the light passes through has its own unique coordinate system depending on its location. The final position is referred to as the GEMS boresight frame for projecting the polarizer angle to the polarizer sheet frame (polarizer angle of 0˚ is aligned toward the eastern direction of GEMS observations (Figure S2)). In order to transform to the coordinate system of the GEMS Boresight frame, the coordinate systems of each component are rotated at each step (Figure S2). As a result, the transformation of LMP to IRP results is similar to a counterclockwise rotation of approximately 90° (Figure 6). The transformation of the coordinate axis involves the following six steps:

1). From the frame of LMP to the spacecraft (GEO-KOMPSAT-2B) body frame

2). to the instrument (GEMS) frame

3). to the reflected sensor output frame

4). to the reflected sensor azimuth/elevation frame

5). to the GEMS boresight frame

6). Projection of the polarization angle to the polarizer sheet frame

In the coordinate transformation, the Stokes component that rotates about each of the x, y, and z axes can be expressed as a quaternion matrix. A quaternion is a mathematical notation for representing the orientation and rotation of an object in three-dimensional space, providing information about its rotation about an arbitrary axis. The amount by which the coordinate system needs to be rotated at each step can be defined through a quaternion multiplication for the entire axis direction via Eq. (3, 4):

$$Q_{axis} = \left[\sin\left(\frac{\alpha}{2}\right) \cdot [x, y, z], \cos\left(\frac{\alpha}{2}\right)\right] \tag{3}$$

$$Q_{step} = Q_x \star Q_y \star Q_z \tag{4}$$

where $Q_{axis}$ is a quaternion matrix for each axis to rotate. α indicates the rotation angle for each axis. As each of the x, y, and z axes is used as a reference axis, it is represented by 1, and the rest are represented by 0 (for example, for rotation in the y-axis direction, the [x,y,z] vector in Eq. (3) becomes [0,1,0]). The total rotational component of each step ($Q_{step}$) is calculated as the product of quaternions in each x, y, and z direction defined in Eq. (3). The symbol "★", indicates quaternion multiplication.

The polarization angle that is defined relative to the LMP, which is the first stage of defining the polarization angle, can be described as a vector form of the quaternion coordinates as follows:

$$V_{LMP} = [cos\chi, sin\chi, 0, 0] \tag{5}$$

Using the value of $Q_{step}$ obtained from Eq. (4), the quaternion vector for polarization angle at each step ($V_{step}^{quat}$) of the transform can be calculated as follows:

$$V_{step}^{quat} = Q_{step}^{-1} \star V_{prev\_step} \star Q_{step} \tag{6}$$

$V_{prev\_step}$ denotes the quaternion vector obtained in the previous step. After the entire steps to the GEMS boresight frame, the last location suggested above, the polarization angle with regards to IRP ($\chi_{IRP}$) is obtained by projecting onto the polarizer sheet frame using the following equation:

$$\chi_{IRP} = \tan^{-1}\left(\frac{V_b^{quat,x}}{V_b^{quat,y}}\right) \tag{7}$$

where, $V_b^{quat,x}$ and $V_b^{quat,y}$ are the x and y axis quaternion vectors defined on the Boresight frame, respectively.

### 3.2 Configuration of Polarization Correction Algorithm

The flow chart and configuration of the GEMS polarization correction algorithm are summarized in Figure 3. The time sequence of the GEMS polarization correction algorithm is divided into a near real-time (NRT) process and a re-process. For NRT, the climatological data on surface Lambertian equivalent reflectivity (LER), surface pressure, and total ozone amount are used as auxiliary inputs. More details for auxiliary data are described below. In the re-process, the GEMS L2 products generated during the NRT process corresponding to the same time period are used as inputs instead of climatological auxiliary data to reduce uncertainties. Since the re-process requires GEMS L2, this paper introduces an approach to NRT as an independent module.

The polarization correction algorithm of GEMS is performed independently for every pixel and involves three processes: 1) determine the pressure of the cloud regions; Starting with the input of GEMS L1B, cloudy regions can be distinguished based on the reflectivity at 477 nm. The cloud top pressure for partially cloudy pixels is derived using the Independent Pixel Approximation (IPA) method, which assumes an albedo of 0.8 for the cloudy regions. This method considers each observed pixel to consist of both cloudless surface and cloudy regions within a plane-parallel atmosphere, and radiative transfer occurs solely in the vertical direction (Choi et al., 2021). 2) calculate the Stokes parameters Q and U from the LUTs; In the next step, values are derived for the Stokes parameters Q and U for the wavelength of each pixel from the LUTs, which is pre-simulated using VLIDORT for various geometries, surface albedos, surface pressures, and trace gases. More details about the LUTs are provided in the next section. 3) the main process of polarization correction. Finally, the polarization correction algorithm is performed using these input parameters. The final result is the radiance which has been corrected for the polarization effect.

### 3.3 Construction of Look-Up Table

Typically, polarization in forward-model radiative transfer simulations must be addressed due to the lack of computational speed and resources. There is a computational time limit to run the RTM in real-time and to correct the polarization effect of all pixels. Therefore, in this study, the efficiency was improved by creating LUTs according to various atmospheric conditions. For this purpose, the LUT for several parameters affecting the polarization degree was prepared. The Stokes parameters ($I$, $Q$, and $U$) were pre-calculated for all wavelengths using VLIDORT as a function of the solar zenith angle (SZA), viewing zenith angle (VZA), relative azimuth angle (RAA), albedo, surface pressure, and ozone profiles. VLIDORT is a discrete ordinate radiative transfer model that treats the multiple scattering. It also includes a feature for simultaneous linearization, enabling the computation of both upwelling or downwelling radiance and analytic Jacobian (not used Jacobian in this study) in multi-layer atmosphere. Unlike typical linearized radiative transfer models, VLIDORT can take polarization into account by generating output for the entire Stokes vector parameters [$I$, $Q$, $U$, and $V$]. Also, Choi et al. (2020) showed that simulation results using VLIDORT are in good agreement with the Stokes fraction (Q/I) measured by GOME-2 PMD over both clear and cloudy conditions. In this study, we use VLIDORT v2.7 in vector mode with 16 discrete ordinate streams. These calculations were performed for the GEMS spectral range (300−500 nm) with a spectral sampling of 0.2 nm. The LUTs contain seven nodes of SZA, seven nodes of VZA, and seven nodes of RAA. The albedo and surface pressures were calculated for five and seven nodes, respectively. The details of the parameters and nodes are summarized in Table 3. The atmospheric conditions (temperature, water vapor, and trace gases) were adopted from the Air Force Geophysics Laboratory (AFGL) atmospheric constituent databases for the United States standard atmosphere 1976 (US76 atmosphere; Anderson et al., 1986), taking into account the absorption of $O_3$, $NO_2$, $SO_2$, HCHO, and $O_2$-$O_2$. The ozone profiles were constructed based on TOMS V8 climatology (Barthia and Wellemeyer, 2002; Wellemeyer et al., 2004). These profiles were classified as low-latitude (L) or mid-latitude (M), depending on the total amount of ozone. The simulation was conducted for a Rayleigh atmosphere.

Due to the slit function, the signal observed by the satellite is affected by the nearby wavelengths, not the signal of a monochromatic wavelength. Consequently, the calculated radiances for monochromatic wavelengths of VLIDORT were convolved by applying a slit function. The convolved spectrum can be expressed as follows:


$$I = P \otimes \tilde{I} \tag{5}$$

where $I$ denotes a convolved spectrum associated with a measurement spectrum from the satellite, $\tilde{I}$ is a high-resolution input spectrum that is calculated monochromatically using VLIDORT, and $P$ is the bandpass function of the instrument, which is
assumed to be a Gaussian slit function with 0.6 nm of the full width at half maximum (FWHM). When sampling the convolved radiance spectrum with 0.2 nm intervals, the radiometric accuracy is affected by the spectral resolution of the reference spectrum (resolution of 0.01 nm) used for convolution. It is necessary to have a high resolution of the reference spectrum to avoid an under-sampling effect due to Nyquist sampling (Chance and Kurucz, 2010).

Figure 4 shows the change in polarization error (determined as $(I_{obs} - I_{true}) / I_{true} \times 100\%$) of the GEMS as a function of
wavelength according to the variation of the six parameters constituting the LUT. $I_{true}$ and $I_{obs}$ denote the simulated radiance without errors and the observed radiance with polarization errors due to the atmosphere and instrument, respectively. The basic atmospheric conditions are set to the general state, M325 (mid-latitude with total ozone of 325 DU) for ozone, 30° of SZA, 30° of VZA, 90° of RAA, 0.05 of surface albedo, and 1013.25 hPa of surface pressure. The simulation was performed by varying each parameter. The polarization errors are most sensitive to changes in geometry (SZA, VZA, and RAA), followed
by albedo, surface pressure, and ozone. The polarization error caused by changes in total ozone is less than those caused by other changes. If the GEMS makes observations with an SZA or VZA of less than 70°, the radiance errors due to instrument polarization sensitivity can approach 2% or higher if polarization correction is not applied.

### 3.3 Climatological Input Data

### 3.3.1 Total Ozone Amount

Ozone plays a crucial role in the atmosphere as a strong absorber in the ultraviolet region, and it was confirmed that absorption by ozone alters the radiance in the Hartley and Huggins bands, which is associated with a change in the degree of linear polarization (Choi et al., 2020). Therefore, the amount of ozone is an effective parameter for analyzing the influence of polarization in the UV region.
In this study, the total ozone climatological data covering the GEMS observation domain were generated using the total column ozone L2 product (short name: OMTO3, http://doi.org/10.5067/Aura/OMI/DATA2024), which is retrieved by OMI to consider the seasonal and spatial variability of total column ozone. This L2 OMTO3 product is based on the TOMS v8 algorithm by the National Aeronautics and Space Administration (NASA), which uses radiance at 317.5 and 331.2 nm (Bhartia and

Wellemeyer, 2002). Each file OMTO3 data product consists of a 1-orbit swath with a spatial resolution of $13 \times 24$ km$^2$ at

nadir. Considering that the spatial distribution of the total amount of ozone does not vary rapidly, the climatological data were generated for a grid of $1° \times 1°$ for latitude (10°S−60°N) and longitude (50°E−170°E), which cover the GEMS field of view (FOV). From 2005 to 2017, OMTO3 was used for a total of 13 years. OMTO3 data of each pixel overpassed through each grid location were integrated and averaged by month.

Figure 5 shows the monthly distribution of the generated total ozone climatological data. From this figure, the pattern of the

annual cycle of the total ozone is evident. In annual ozone distribution, the total amount of ozone in tropical regions is smaller than in mid-latitude regions, and the seasonal variations in mid-latitude regions are well expressed. This periodic pattern is controlled by the balance between the transport and photochemical loss of ozone. The amount of ozone increases in winter when transport is predominant and decreases in summer when transport dwindles and photochemical loss dominates (Andrews et al., 1987). The distribution of the total amount of ozone varies clearly with latitude. The zonal average of the total ozone

increases rapidly in mid-latitudes over 30°N.

Furthermore, due to the high topographical altitude of the Tibetan Plateau, the low total amount of ozone is well represented. The maximum and minimum value of the total ozone was 475 DU and 229 DU in February and January, respectively. The closest grid of total ozone climatological data was collocated to each pixel of GEMS L1B, and the total ozone amount corresponding to the month in which the GEMS L1B was observed was used as an input value for the algorithm.

**3.3.2 Lambertian Equivalent Reflectivity**

Surface reflectivity is defined as the ratio of the incident sunlight to the sunlight that is reflected from the Earth's surface, which differs depending on the state of the surface, its constituent components, the direction of light propagation, and the light's wavelength. Therefore, since the surface reflectivity varies depending on the wavelength and the season, and its properties are very different according to the spatial location with different land cover types, it is essential to apply a

wavelength-dependent reflectance that considered the variation of characteristics of the surface independently for each pixel. Two theoretical concepts can be considered when analyzing surface reflectivity: a bidirectional reflectance distribution function (BRDF) with directional dependence and an LER that assumes no anisotropy of reflectivity. The BRDF can explain the comprehensive characteristics of surface reflectivity. Nevertheless, LER was used in this study because insufficient BRDF climatological data are available for the radiative transfer model. Many previous studies attempted to extract LER information

from satellite observations (Kaufman et al., 1997; Schaaf et al., 2002; Hsu et al., 2004 and 2006). In this study, GOME-2 surface LER climatology data (Tilstra et al., 2017) were used, which cover the GEMS spectral range among the existing LER databases. The GOME-2 surface LER climatology is constructed based on the observation from the Main Science Channel (MSC) with a pixel resolution of $80 \times 40$ km$^2$. It is provided as a monthly averaged LER for 21 wavelengths (from 335 to 772 nm). Among them, 11 wavelengths are included in the GEMS spectral region (300–500 nm). The spatial resolution of the

GOME-2 surface LER is $1° \times 1°$, and it was interpolated to $0.25° \times 0.25°$ for a more accurate surface representation for the coastlines or snow-covered mountainous areas. The GOME-2 surface LER that was used as the input data for the polarization

correction algorithm was derived at each pixel position of GEMS L1B. The latitude and longitude point of the GOME-2 LER closest to the GEMS pixel was selected and the LER spectrum corresponding to the month in which the GEMS L1B was observed was obtained. The obtained LER spectrum for the 11 wavelengths (335, 340, 354, 367, 380, 388, 416, 425, 440, 463, and 494 nm) was interpolated to the GEMS wavelengths.

### 3.3.3 Terrain Height

Terrain height is one of the parameters associated with optical thickness while sunlight passes through the atmosphere and is reflected by the Earth's surface. The optical path length from a satellite to the Earth's surface strongly depends on the atmospheric pressure along the propagation path. For this reason, terrain height information is included in the L1B of various satellites.

In this work, the Earth TOPOgraphy (ETOPO)-2 dataset (NOAA National Geophysical Data Center, 2006) was used to obtain the terrain height information within the GEMS observation domain. ETOPO-2 provides altitude information on the Earth's crust and was produced using many digital databases of the seafloor and land elevations on a 2 arc-minute latitude/longitude grid. Many datasets were used to produce ETOPO-2, such as satellite altimetry observations, shipboard echo sounding, Digital Bathymetric Data Base Variable resolution (DBDB-V; Sandy, 1996) data, and Global Land One-kilometer Base Elevation (GLOBE; House, 2004) project data, the latter of which include a Digital Elevation Model (DEM). The coverage of ETOPO-2 is from 90°S to 90°N latitude and from 180°W to 180°E longitude. In order to consider only the surface altitude, submarine regions with a negative altitude value were assigned an altitude of zero. The terrain height is collocated with the grid position closest to each pixel of GEMS L1B in the same way as LER, to adapt to spatial locations that depend on the GEMS observation schedule. The generated terrain heights were converted into surface pressure to be utilized in the RTM. Altitude can be easily converted to pressure using the following barometric formula by assuming that all pressure is hydrostatic:

$$P = P_0 exp^{(-\frac{z}{H})} \tag{5}$$

where $P$ is the surface pressure, $P_0$ is the pressure at sea level (1013.25 hPa), $z$ is the surface altitude, and the scale height ($H$) is assumed to be 8 km. Actual surface pressure is heterogeneous and varies over time. Nevertheless, since it is difficult to determine the actual value at every moment, there is a limitation to using the numerical forecasting data in NRT. The polarization correction algorithm is insensitive to the reflecting pressure under clear sky conditions and the use of a terrain height pressure results in a negligible error. Therefore, it is useful to estimate the surface pressure using the terrain height.

## 4 Assessment of Polarization Correction Algorithm

### 4.1 Performance test using Synthetic data

The performance of the polarization correction algorithm for the polarization effect due to the GEMS instrument and the atmosphere was evaluated using synthetic data. The simulated data ($I_{true}$) of the actual atmosphere generated by the RTM were converted into synthetic data assumed to have been observed by the satellite ($I_{obs}$) by considering the inherent polarization characteristics of the GEMS instrument. Then, the GEMS polarization correction algorithm was performed. Finally, the proposed algorithm's polarization-corrected radiance ($I_{polcor}$) was compared with the actual value ($I_{true}$).

The synthetic data were generated using VLIDORT for the GEMS domain, including aerosols and clouds, as well as absorption by atmospheric gaseous components. The contents of the simulation data, the geometric information that is determined as a function of the satellite-sun geometry (SZA, VZA, and RAA), and the Stokes parameters (Q and U components) are depicted in Figure S3 and S4. Note that the radiance in cloudy regions is higher than that for a clear sky (Figure S5a), whereas the Q and U components are lower. The polarization-related parameters used for the polarization correction of the synthetic data are shown in Figure 6. Figures 6a–c present the spatial distribution of the polarization angles for LMP, the polarization angles for IRP, and the degree of linear polarization, respectively. The polarization angle converted to the coordinate reference frame of IRP corresponds to the effect of rotating the polarization angle of the LMP by approximately 90° anticlockwise. As clouds play a role in depolarizing light, DoLP is smaller for the cloud region than for the clear sky.

Figure 7 shows the spatial distribution of the relative error (which denotes the polarization error) before and after polarization correction for several representative wavelengths (349.6, 432.0, 454.6, and 494.8 nm) that exhibit sharp curvatures in PF (as shown in Fig. 2). The spatial distribution characteristics of the polarization errors are influenced by the PF and the PA and vary depending on the observation geometry and wavelength. For these 4 wavelengths, the maximum range for the polarization error of the radiance before polarization correction is at most ± 0.05% (Even not shown here for all wavelengths, a polarization error of up to 0.1% occurs depending on the wavelength). After performing the polarization correction algorithm, the polarization error was reduced in all wavelengths and regions. The histogram of the polarization error for the entire domain of each wavelength (Figure 8), the mean of the polarization error, and the FWHM assuming Gaussian distribution are summarized in Table 4. After the polarization correction, the mean value of the polarization error becomes close to zero for all wavelengths. The FWHM, which can indicate the degree of spread of the polarization error, was reduced by more than half (Table 4). In particular, at 331 nm and 388 nm, where the polarization error is relatively large, the FWHM decreased by 4 and 3.5 times, respectively. Figure 9 shows the polarization error and standard deviation before and after polarization correction for the clear sky and cloud areas in the entire domain. Overall, the curvature of polarization error as a function of wavelength is determined by the PF, and the sign is determined by the PA. The polarization error before polarization correction in the cloud region is lower than that in the clear sky region because the DoLP is decreased by the cloud attenuating the polarization. In both regions, the polarization error is reduced to almost zero for all wavelengths after polarization correction.

However, even after the polarization correction is performed, a slight polarization error remains. These residual errors are related to the interpolation method using the LUT to derive the polarization parameters. As depicted in Figure 10 and S4, the approach of the LUT method presented in this study yields results that are very similar to those obtained by online calculations of the RTM, as well as in terms of sign and spectral features of the Stokes parameters, for a given geometry. However, the LUT method still exhibits very small discrepancies in magnitude for geometries that vary at the decimal level, resulting in imperfect matching. For example, the difference between the average of 15% and 24% for Q and U shown in Figure 10 causes difference of 21% (0.003) in DoLP and 2% (0.02) in polarization angle. However, even if the relative error between each variable is large, the absolute value is very small, so this effect remains only with a polarization error of about 0.005%. In addition, the errors can arise because we do not consider the aerosol influence and assume the cloud to be Lambertian. According to Choi et al. (2020), the degree of polarization attenuation varies depending on the Aerosol Optical Depth (AOD) and aerosol height. According to Choi et al., 2021, the degree of polarization attenuation varies depending on the aerosol optical depth and aerosol height. The top height of aerosol at 1.8 km and 3.6 km decreases by ~15% and ~18% compared to the DoLP of Rayleigh atmosphere, respectively. Additionally, for the same aerosol loading height, the DoLP decreased from ~2% to 25% when AOD varies from 0.1 to 2.0. This suggests that even if aerosol influence is inherent in the cloud processing process, polarization error may be overcorrected if corrected for the clear sky without considering aerosols. The difference in DoLP between the assumption for Mie clouds and Lambertian clouds is small, and for very high-altitude clouds (above 10 km) Mie clouds tend to attenuate polarization slightly more than Lambertian clouds. Rather than how the cloud is treated, an important factor in polarization, as with aerosols, is the cloud top height (cloud surface pressure). The residual polarization error in the cloud region is higher than in the clear sky. As it is difficult to calculate the correct polarization states for cloud regions that are not a Rayleigh atmosphere, there remains room for further improvement in cloud regions. These are discussed in the Discussion section in more detail. Lastly, the point to note is that the influence of the spectral features at some wavelengths caused by the coating of the Schmidt mirror in the PF feature of the GEMS was clearly revealed before polarization correction and then corrected after polarization correction.

## 4.2 Application to GEMS observation data

The performance of the GEMS polarization correction algorithm was evaluated using synthetic data in the previous section. The GEMS currently in operation is scheduled so that the scan region varies according to the sun's position. However, it is difficult to accurately grasp the diurnal variation of polarization over time in the same observed domain if the scan area fluctuates. Thus, in order to better understand the diurnal variation of the polarization error, we selected and analyzed a specially scheduled date (25th July 2020) to measure the same domain for a whole day among the in-orbit test (IOT) periods. Figures 11 and 12 show the corrected polarization error (in other words, the implicated polarization error in observed L1B) during the day (00–05 UTC) by applying the polarization correction algorithm to actual experimental data that was obtained after the GEMS was launched. The degree of distribution of polarization error gradually decreases over time from the maximum in the morning (00 UTC, ± 0.5 %) to the minimum around noon (02–03 UTC, ± 0.15%) and then increases again

until before sunset. (05 UTC, ± 0.45%). The polarization error is three times higher around dawn/sunset compared to around noon, when it is the smallest. As noticed in Fig. 4, this diurnal variation is greatly affected by the change of the SZA (the larger the SZA, the larger the polarization error, and the smaller the SZA, the smaller the polarization error). These non-constant variation in polarization error in a day can also affect the performance of L2 products, preventing accurate retrieval. Therefore, it is critical to apply the polarization correction considering the time and location, in order to obtain accurate and reliable

measurements, and it is necessary to analyze its effect on L2 products in future.

## 5 Discussion and conclusion

The UV-Vis sensor is sensitive to the polarization of incident light, and the polarization sensitivity of a satellite instrument is one of the key characteristics for securing radiometric accuracy. In order to improve the radiation accuracy by reducing the polarization error included in the radiance spectrum observed from the GEMS, a LUT-based GEMS polarization correction

algorithm considering both the polarization characteristics of the instrument and of the atmosphere simultaneously was developed. The performance of the developed GEMS polarization correction algorithm was evaluated by using synthetic data. In the GEMS observation domain, the polarization errors are larger in the clear sky than those in cloudy regions because clouds attenuate the polarization of the atmosphere. Then the polarization error becomes very small because the DoLP of the incident light is reduced. After applying the polarization correction, the polarization errors were reduced to zero for almost all

wavelengths, and the high peaks of PF that occur at specific wavelengths were almost corrected. In addition, it was demonstrated that the spatial distribution of the polarization error varies via the sun's location, and the largest polarization error occurs at sunrise/sunset time by analyzing the actual GEMS observation data.

However, some limitations and problems need to be improved in correcting the errors due to polarization in the radiance spectrum, and these uncertainties deserve to be identified and quantified. The error factors considered in the polarization

correction process can be categorized into two groups: errors that may arise during the Radiative Transfer Model (RTM) simulation using VLIDORT and errors associated with the characteristics of GEMS. The most basic error is the error in the radiative transfer calculation of VLIDORT to simulate the TOA radiance. Castellanos et al. (2018) demonstrated that the simulation error was around 0.1% by comparing the results of other polarization radiation transfer models for the atmospheres containing various configurations of Rayleigh scattering, aerosol scattering, and molecular absorption of VLIDORT.

Besides, even after polarization correction, there are very slight residual errors, which are due to the uncertainty of applying the linear interpolation based on the LUT, which is constructed at regular intervals. This is because the Stokes parameters that describe the polarization state in the actual atmosphere do not vary perfectly linearly according to each atmospheric state. The error due to linear interpolation tends to increase as the angles of SZA, VZA, and RAA increase. As noted in Section 4.1, the polarization errors in the stokes components caused by the LUT approach can be caused as small as two or three decimal

places. This problem can be improved by optimizing the interval of each parameter and the number of nodes that can minimize the error due to linear interpolation. The possible errors in this comprehensive radiative simulation process cannot be ignored, but they are small.

The slight polarization errors which remain even after polarization correction as shown in Fig. 7 is mostly like to due to uncertainties the aerosol contained in the synthesis data is not considered or in the process of estimating the surface pressure of the cloud regions. By not considering aerosols over cloud-free regions, due to the relatively less effect on the degree of atmospheric polarization than clouds, the effects of areas with strong dust or aerosol plumes are not fully accounted. In the presence of aerosols in the atmosphere, DOLP of atmosphere varies with the height of the aerosol layer, and as the AOD increases, DOLP of atmosphere gradually decreases compare to Rayleigh atmospheric condition. For example, the presence of 1.0 AOD compared to the clear sky results in a reduction of about 20% of DOLP of atmosphere at 432 nm (Choi et al., 2020). This means that if polarization correction is performed by assuming an aerosol-containing pixel as a clear sky condition, over-correction may be performed. In addition, there are errors introduced by assuming cloud as Lambertian without considering scattering by cloud particles in our derivation of cloud parameters. In particular, the polarization effects of anisotropic ice particles present in the cirrus or upper cloud are overlooked. Likewise, when the surface pressure or cloud top pressure are estimated to be lower/higher than actual state, the polarization effect can be overcorrected/undercorrected. The next point is, the spectral calibration issue that related to the characteristics of GEMS. Since the polarization characteristics (PF and PA) is a function of wavelength, and the wavelength registration of each Earth scene relies on radiance spectral structure can be affected. Therefore, it is important to apply the polarization correction at the correct wavelength position. The polarization correction algorithm is assumed that the spectral calibration perfectly executed in the previous step of L1B. If the allocation of the wavelength is not clear, the polarization error might be inaccurately corrected in the vicinity of non-continuous polarization error regions, particularly where the polarization fraction exhibits the steepest increase. Another limitation is the lack of characterizing polarization sensitivity in the spatial variation (North/South cross-track direction). By performing a polarization test of GEMS on the ground, the polarization sensitivity was inferred for only one SMA position (corresponding to the center of the CCD as the north-south direction). Therefore, there is insufficient information on the variability of PF and PA in the north-south direction. Even if polarization correction is performed, undefined polarization errors can be included in the observation. According to the BATC model, the ratio of the polarization factor for the north/south directions can increase by up to 6 times within 350 to 400 nm. Assuming this, polarization errors of Fig. 7 of up to 0.4% occurs within 350 to 400 nm, and even after polarization correction with the current proposed algorithm, the polarization error of 0.3% remains. Thus, there is room for this to be solved in the future by introducing an additional method for deriving the spatial variation of polarization characteristics (For instance, a scaling method by modeling a change in polarization sensitivity according to a change in scan mirror angle).

Besides, the impact of spatial-temporal polarization correction effects on the retrieval performance of L2 products should be assessed. In particular, the wavelength-dependent variability of polarization errors in GEMS, characterized by a jagged curve shape at relatively short wavelengths (<350 nm) and a sharp increase at specific wavelength ranges, can potentially impact on the performance of L2 retrieval algorithms utilizing this wavelength range, such as $O_3$ (Baek et al., 2022), HCHO (Kown et al., 2019), and aerosols (Kim et al., 2018; Go et al., 2020). Therefore, by analyzing the impact of polarization (e.g., the presence

or absence of polarization correction) on L2 product retrieval, the improvements in the accuracy of the L2 products can be expected in the future.

## Data availability

GEMS measurement data are available on request from the National Institute of Environmental Research (NIER) Environmental Satellite Center (ESC) (https://nesc.nier.go.kr; NIER, 2023)

## Author contributions

HC, XL, and UK conceptualized the study; HC, DK helped algorithm modification implementation; Supervision was the JK, MA, and KL; DL and KM assisted in providing the GEMS data; HC drafted the original manuscript; All the co-authors provided comments and contributed to editing the manuscript and figures.

## Competing interests

The authors declare that they have no conflict of interest.

## Disclaimer

Publisher's note: Copernicus Publications remains neutral with regard to jurisdictional claims made in the text, published maps, institutional affiliations, or any other geographical representation in this paper. While Copernicus Publications makes every effort to include appropriate place names, the final responsibility lies with the authors.

## Special issue statement

This article is part of the special issue "GEMS: first year in operation (AMT/ACP inter-journal SI)". It is not associated with a conference.

## Acknowledgements

We would like to thanks to Dr. David Flittner of NASA Langley Research Center for providing the useful comments and interpretation.

## Financial support

This research was supported by the National Research Foundation of Korea (NRF) grant funded by the Korean government (MSIT) (no. 2020R1A2C3003774). This research was supported by the Korea Ministry of Environment (MOE) through the "Public Technology Program based on Environmental Policy (2017000160002)".

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

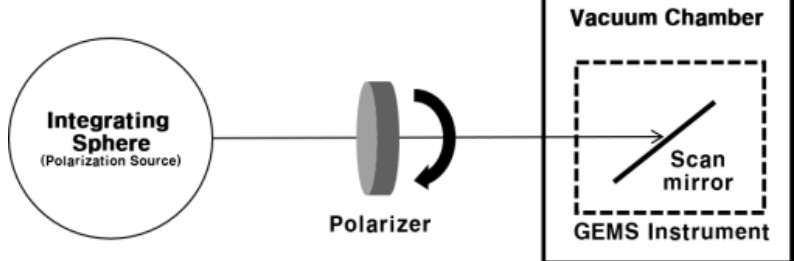

**Figure 1: Symmetric diagram of on-ground polarization test for GEMS.**

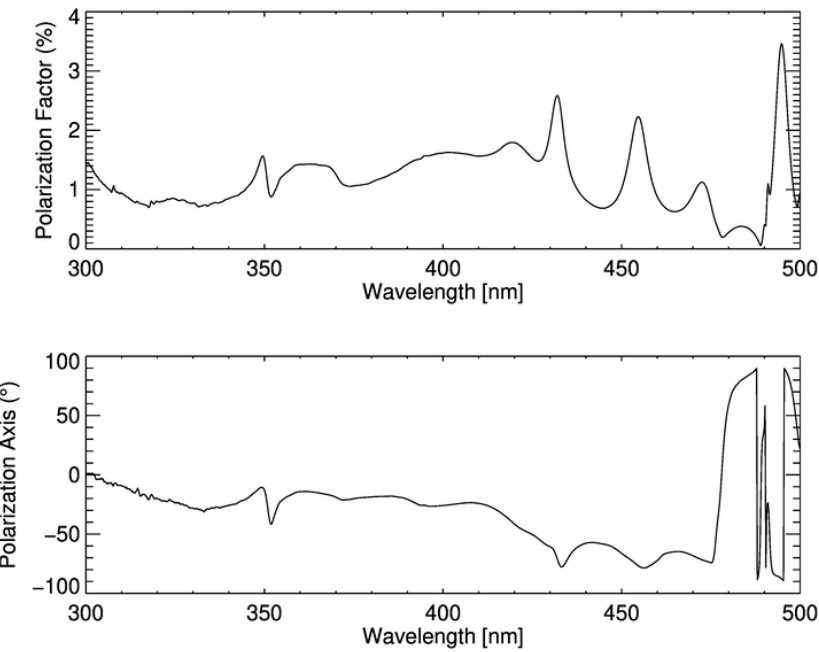


**Figure 2: Polarization factor and polarization axis as a function of wavelength from 300 to 500 nm, derived from the polarization sensitivity test.**

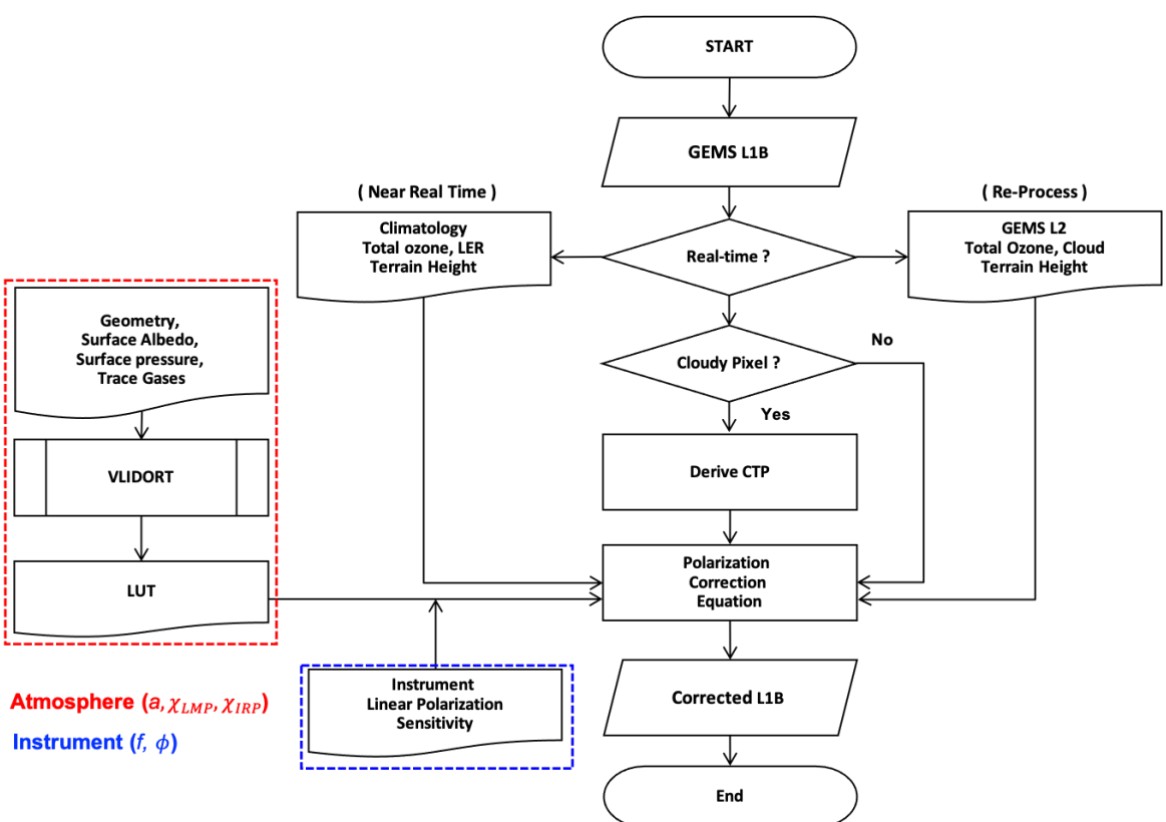

**Figure 3: Schematic diagrams of the structure and the sequence flows of the polarization correction process for GEMS. The red and blue squares with dashed lines represent the atmospheric polarization ($a$; degree of linear polarization, $\chi_{LMP}$; polarization angle for local meridian plane, $\chi_{IRP}$ ; polarization angle for instrument reference plane) and instrument polarization parameter ($f$; polarization factor $\phi$; polarization axes), respectively.**


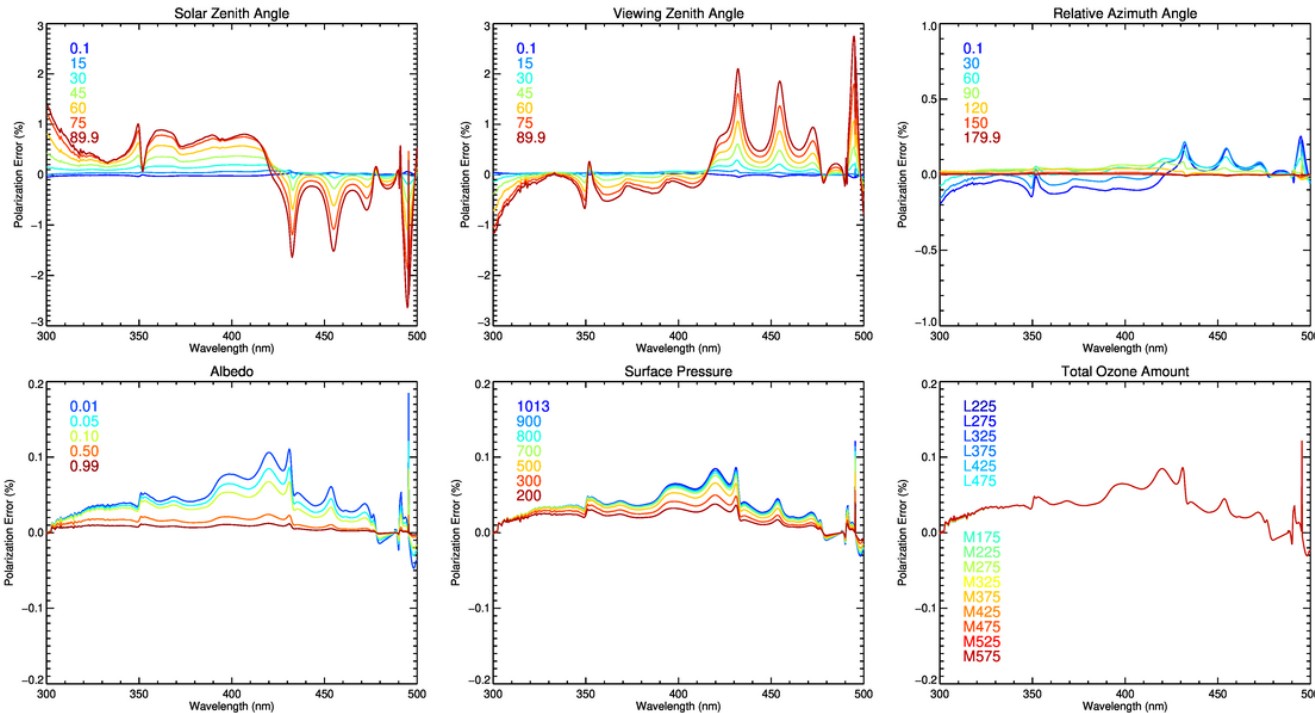

**Figure 4: The results of polarization error sensitivity tests of the influencing factors for polarization. The basic conditions of the simulation are M325 (mid-latitude with total ozone of 325 DU) for ozone, 30° of SZA, 30° of VZA, 90° of RAA, 0.05 of surface Albedo, and 1013.25 hPa of surface pressure. The simulation was performed for each change of a parameter.**

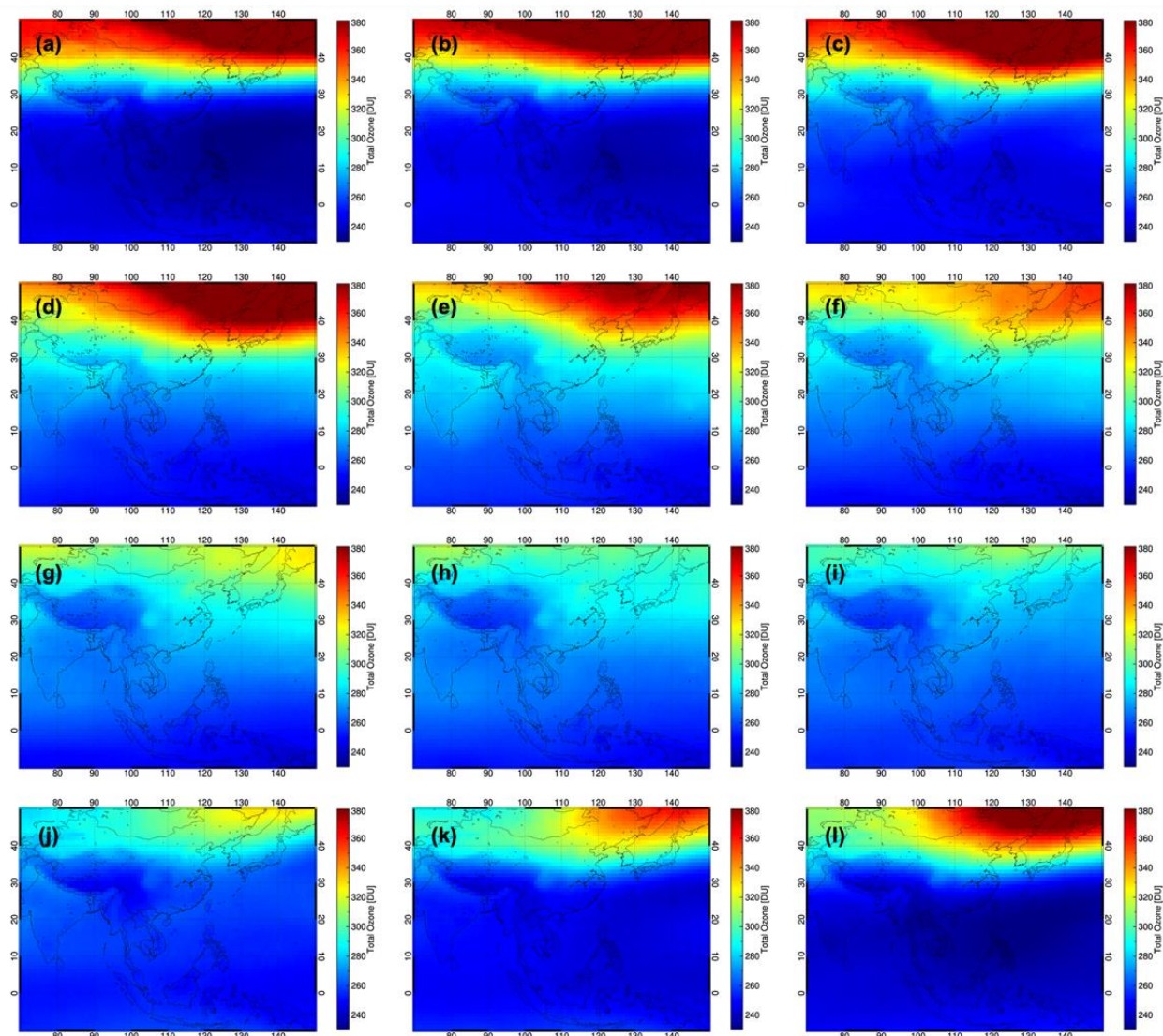

**Figure 5: The distribution of the total column ozone climatological data within the GEMS observation domain. The dataset was created based on the OMI L2 product OMTO3. (a)-(i) represent the annual variation sequentially from January to December.**

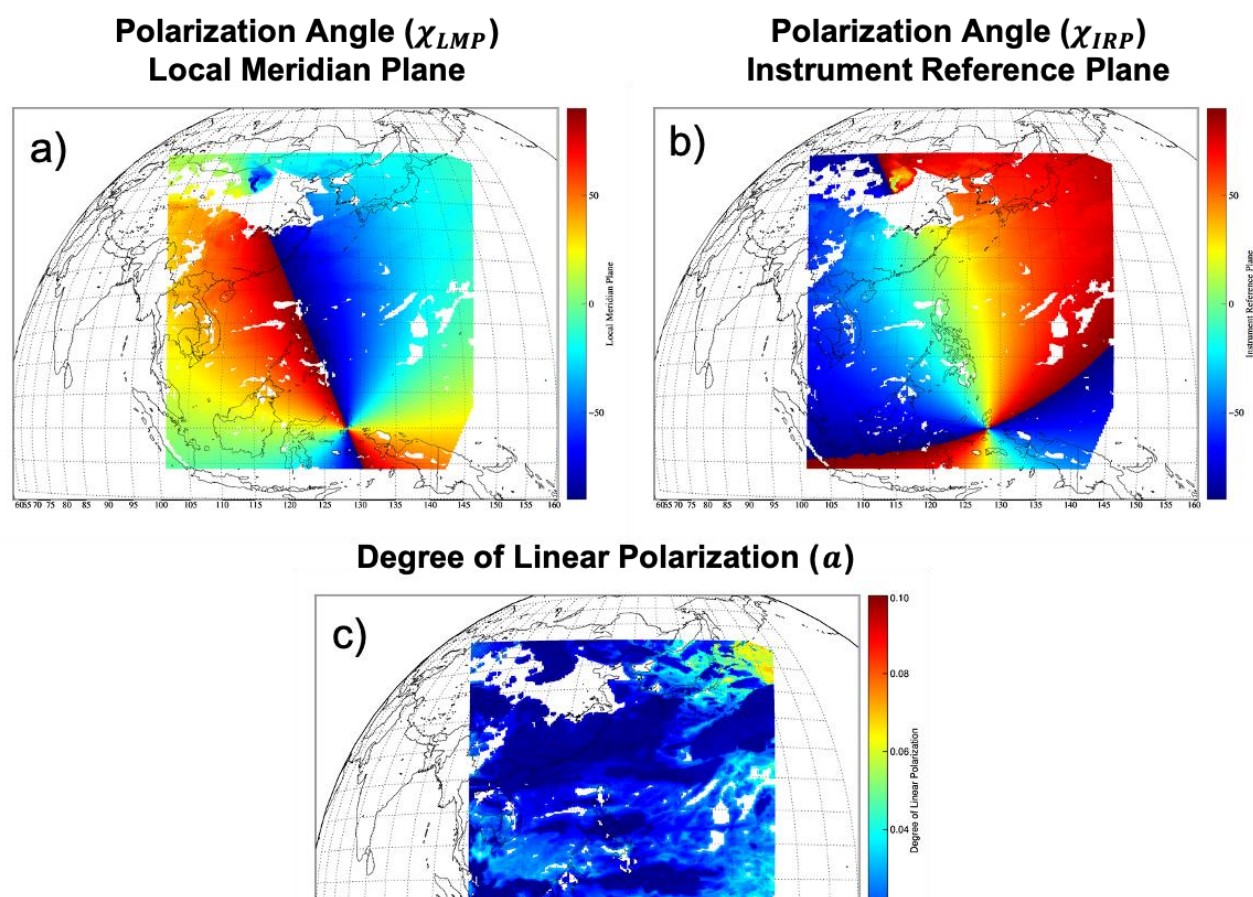

**Figure 6: The derived polarization angle with respect to (a) the Local Meridian Plane (LMP) and (b) the Instrument Reference Plane (IRP). (c) shows the degree of linear polarization at 432 nm. The blank regions indicate areas with poor pixel quality.**


.

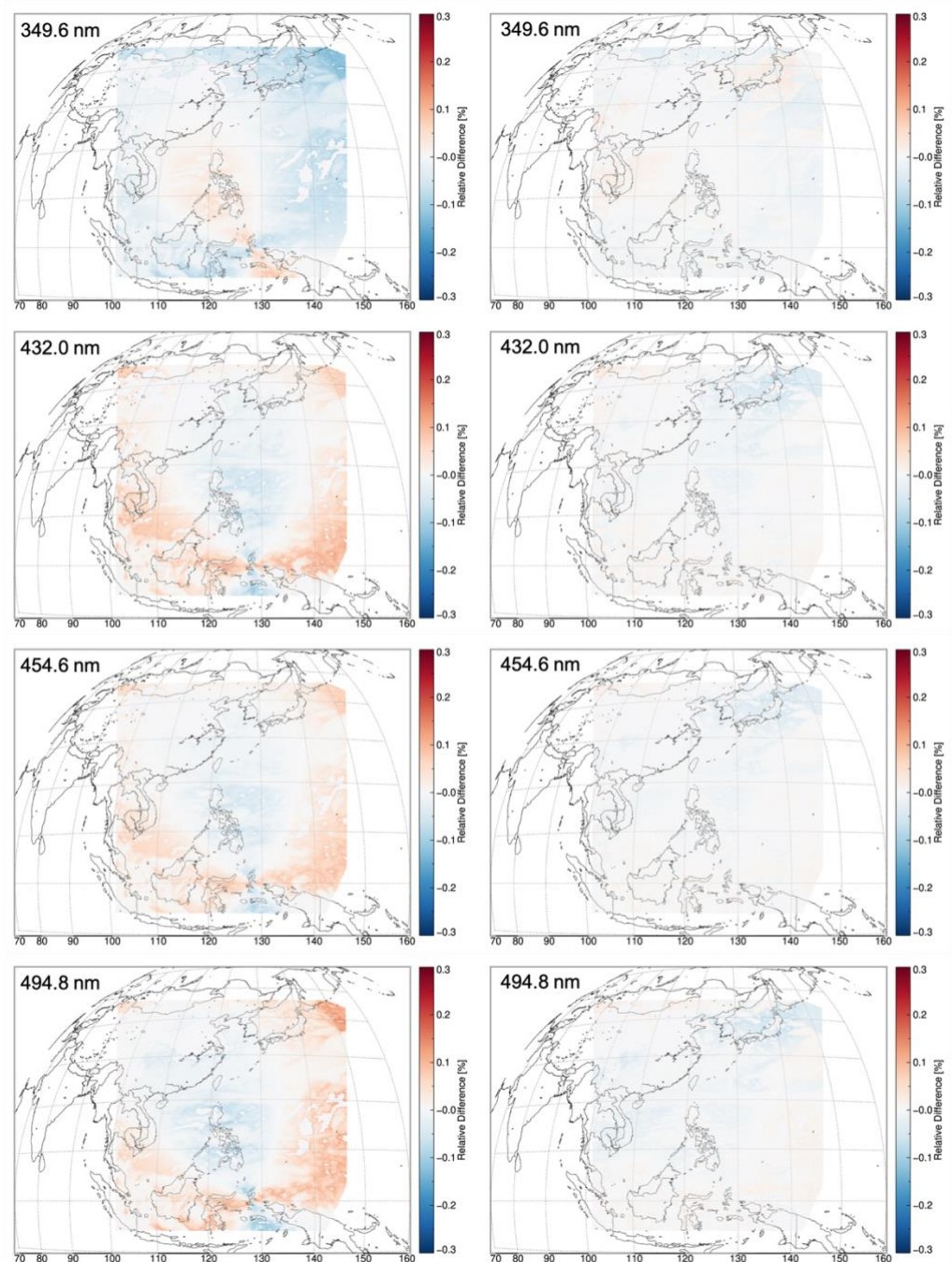

**Figure 7: The spatial distribution of polarization error for before (left panels) and after (right panels) polarization correction in the GEMS observation domain for specific 4 wavelengths (349.6, 432, 454.6, and 494.8 nm).**

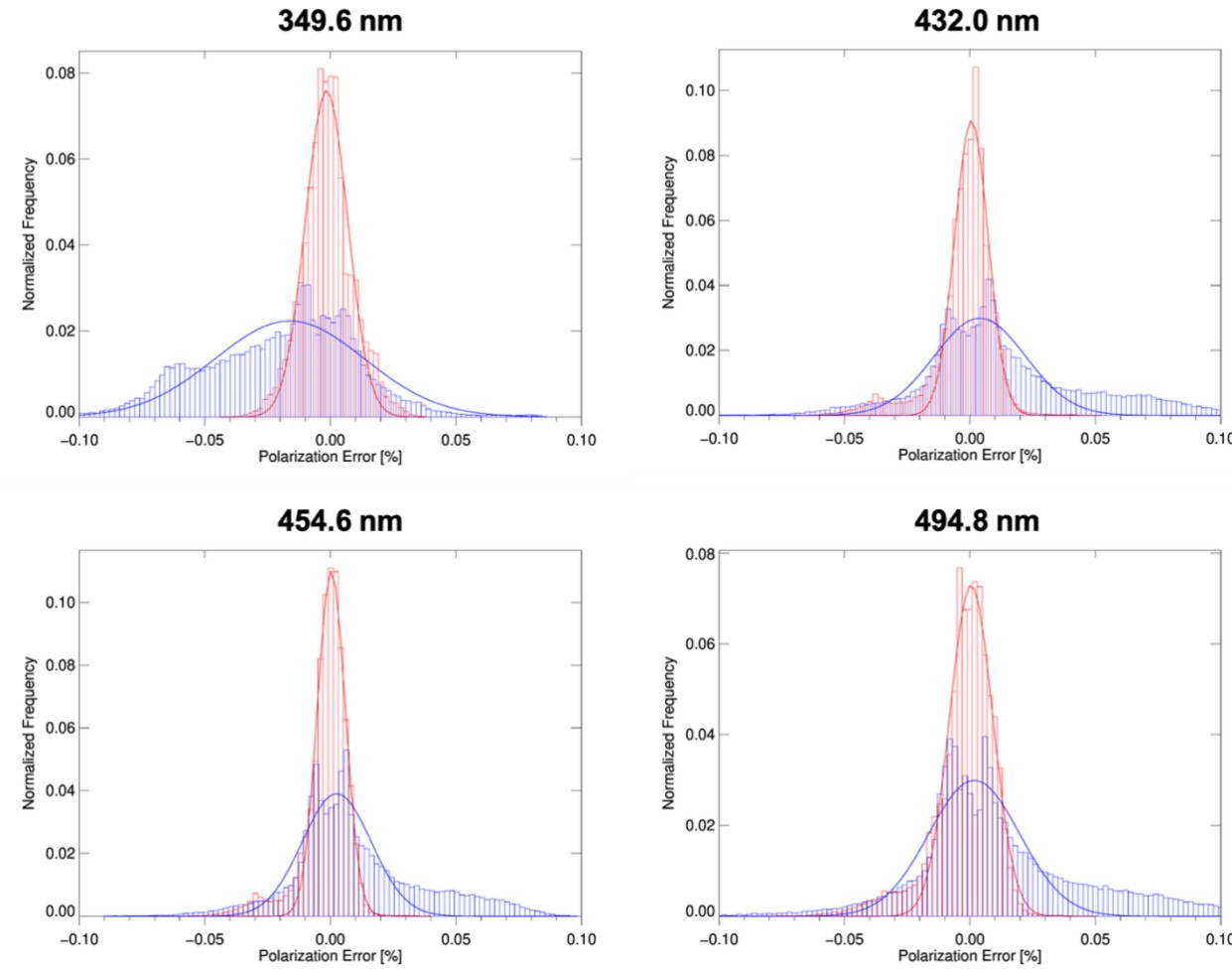


**Figure 8: Histograms of the polarization error before (blue) and after (red) polarization correction.**

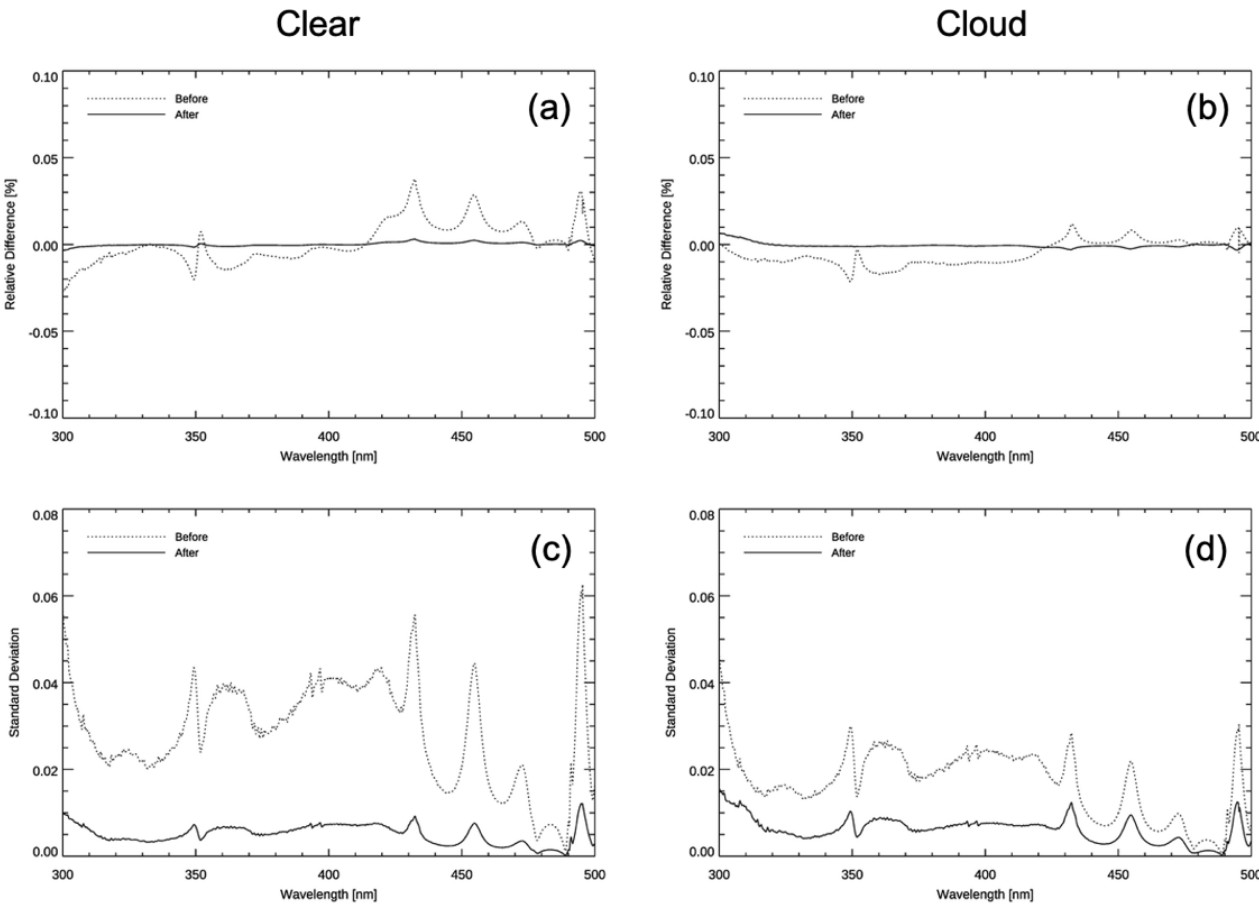

**Figure 9: Comparison of the relative difference (top) and standard deviation (bottom) of radiance before and after polarization correction for clear sky pixels (left panels) and cloudy pixels (right panels)**

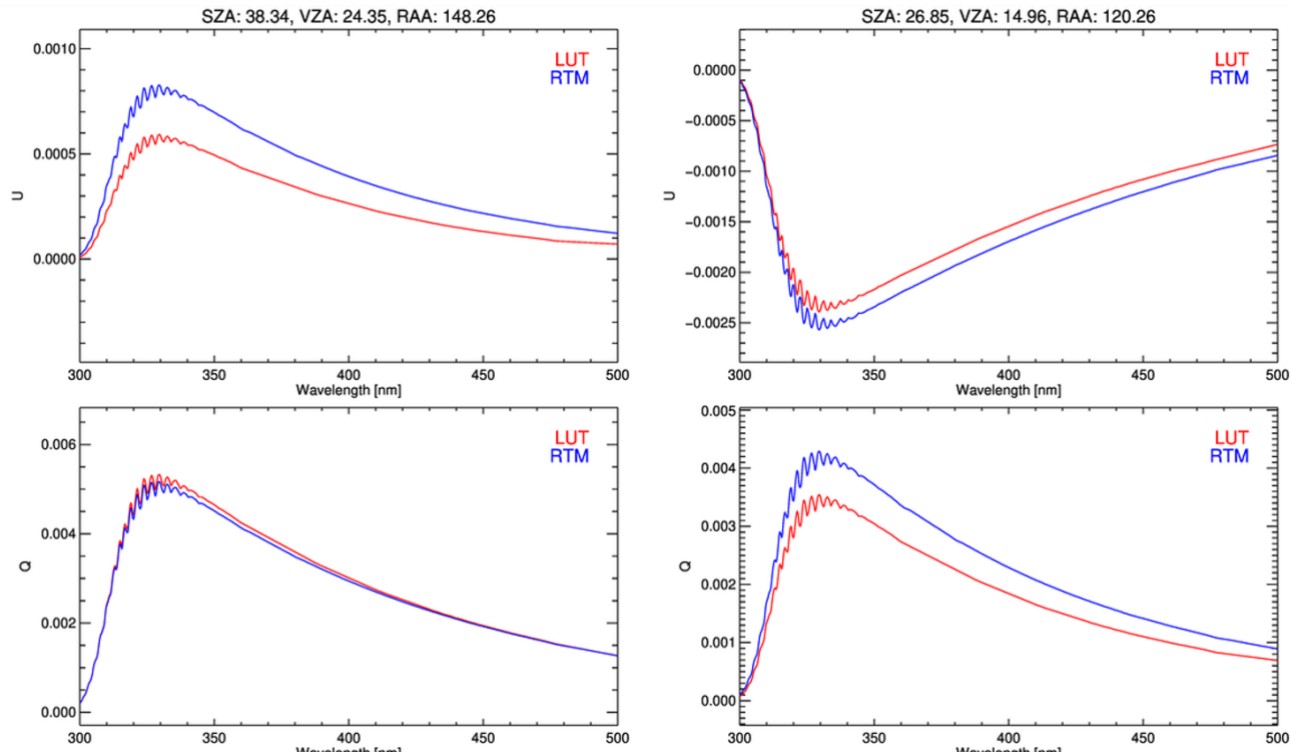

**Figure 10: Comparison of Stokes parameters (Q and U) estimated using the look-up table (LUT) method (red) and on-line calculated synthetic data by RTM (blue) for given geometries (solar zenith angle, viewing zenith angle, and relative azimuth angle) at two different locations.**

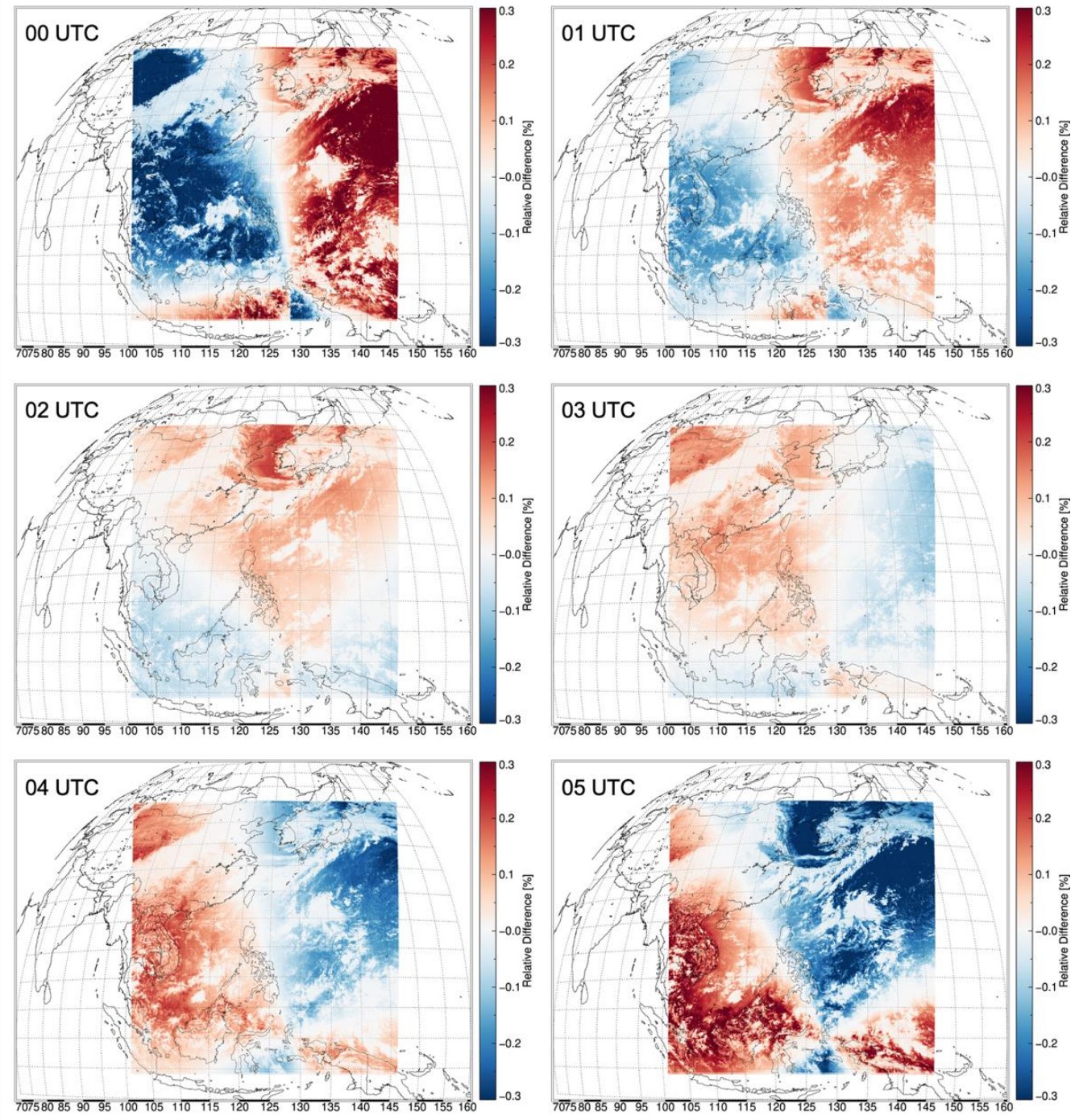

**Figure 11: The diurnal variation of the spatial distribution of the corrected polarization error as precent difference by the polarization correction algorithm for the actual GEMS observation data on July 25, 2021, from 00 to 05 UTC. Note that the observation on that date is during the GEMS in-orbit test (IOT) period which the image navigation and registration (INR) is not completed. The presented wavelength is 432 nm.**

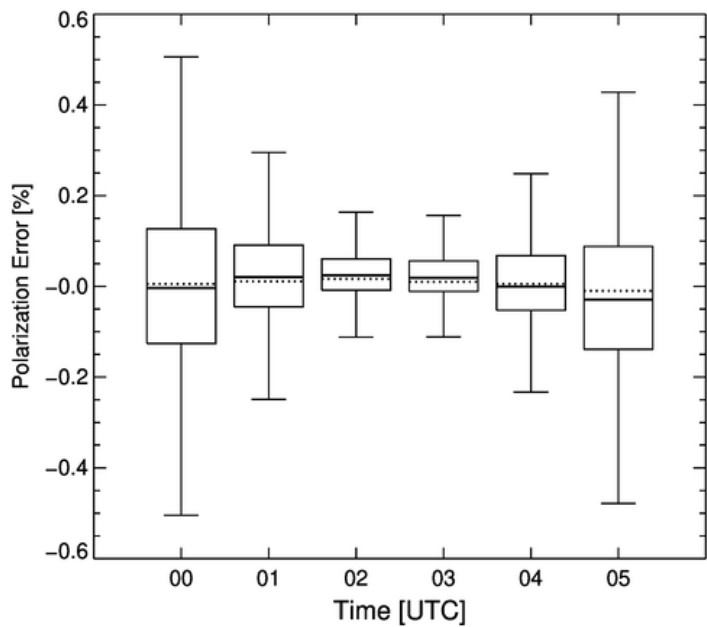

**Figure 12: Box-whisker plot of the diurnal variation (00 to 05 UTC) for the spatial distribution of corrected polarization errors by the polarization correction algorithm is shown in Figure 11. The box encloses the interquartile range (IQR) defined at 25–75 percentiles, and whiskers represent maximum and minimum. The solid and dot lines refer to the mean and median values of the data, respectively.**



**Table 1: Specification of GEMS instrument**

| Parameter | Value |
|---|---|
| Spacecraft | GEO-KOMPSAT-2B |
| Orbit | Geostationary |
| Lifetime | > 10 years |
| Spectral range | 300–500 nm |
| Spectral resolution | 0.6 nm |
| Spectral sampling | 0.2 nm |
| Temporal resolution | 1 hour |
| Spatial resolution | $7 \times 8$ km$^2$ (gases) at Seoul<br>$3.5 \times 8$ km$^2$ (aerosol) at Seoul |
| Field of regard | $> 5000 \times 5000$ km$^2$ (N/S $\times$ E/W)<br>N/S range: 5°S - 45°N<br>E/W range: 75° - 145E |
| Requirement of polarization factor | < 2 % (310–500 nm)<br>(No inflection point within 20 nm range) |


**Table 2: Setup environment for on-ground polarization test of GEMS**

| Configuration Condition | Status |
|---|---|
| Integration Time (msec) | 50 |
| Number of co-adds per image | 60 |
| Number of SMA position | 1 |
| Polarizer start angle (°) | 0 |
| Polarizer end angle (°) | 720 |
| Polarizer angular step size (°) | 5 |
| Number of polarizer step | 145 |
| Number of repeatability test | 10 |


**Table 3: Details of parameters used to construct LUT using VLIDORT**

| Parameter (Unit) | Entries |
|---|---|
| Spectral range (nm) | 300–500 |
| Spectral sampling (nm) | 0.2 |
| Solar Zenith Angle (°) | 0.1, 15, 30, 45, 60, 75, 89.9 |
| Viewing Zenith Angle (°) | 0.1, 15, 30, 45, 60, 75, 89.9 |
| Relative Azimuth Angle (°) | 0.1, 30, 60, 90, 120, 150, 179.9 |
| Surface albedo | 0.01, 0.05, 0.1, 0.5, 0.99 |
| Surface pressure(hPa) | 1013.25, 900, 800, 700, 500, 300, 200 |
| Total ozone amount (DU) | L225, L275, L325, L375, L425, L475, M175, M225, M275, M325, M375, M425, M475, M525, M575 |

Note, L and M indicate low-latitude (<30) and mid-latitude (>30)

**Table 4: Statistical results before and after polarization correction for the selected four wavelengths (349.6, 432.0, 454.6, and 494.9 nm) presented in Figures 7 and 8.**

| Wavelength (nm) | Before / After | | | | |
| --- | --- | --- | --- | --- | --- |
| | mean | median | 16th percentile | 84th percentile | FWHM |
| 349.6 | -0.021 / -0.001 | -0.010 / 0.001 | -0.050 / -0.009 | 0.655 / 0.027 | 0.070 / 0.020 |
| 432.0 | 0.013 / -0.002 | 0.012 / 0.001 | -0.010 / -0.008 | 0.103 / 0.022 | 0.043 / 0.016 |
| 454.6 | 0.010 / -0.002 | 0.009 / 0.001 | -0.008 / -0.006 | 0.083 / 0.015 | 0.032 / 0.013 |
| 494.8 | 0.012 / -0.002 | 0.011 / 0.002 | -0.011 / -0.010 | 0.118 / 0.020 | 0.043 / 0.020 |