# Peer review of "Geostationary Environment Monitoring Spectrometer (GEMS) polarization characteristics and correction algorithm"

_Atmospheric Measurement Techniques, 2023_

## Author Comment (AC1)

**Response to Reviewer #1**

We appreciate your very meaningful comments.
It gave us a deeper understanding of what we overlooked and didn't take into account, which enriched the manuscript.

Line 45-46 Replace "Before reaching ... passes through …" with "Upon reaching … interacts with …"

→ Revised. We modified the sentence as you suggested.

Line 91 This needs to be rewritten. Maybe the Stokes parameters (I, Q, and U) for various atmospheric conditions were calculated and the DoLPs are arranged in a LUT.

→ DOLPs are not directly included of LUTs as parameter. Instead, the sentence has been improved by replacing "comprised of LUT" with "included of LUT" in the original wording.

Line 126 The acronym SMA need to be expanded. Is it Scan Mechanism Assembly? Scan Mirror Angle? Also, sometimes it is used as "SMA Angle: and other times as "SMA Position" or "angle at which the SMA is located".

→ Revised, it denotes "Scan Mirror Assembly (SMA)". I wrote down the full word of the SMA acronym.

Line 128 Linear Polarization Sensitivity or Polarization Factor? The term LPS is introduced in Line 89 but is never used. The term PF is used often.

→ Revised, in order to avoid confusing, we have unified the representation with Polarization Factor (PF). LPS and PF are often used interchangeably, but to avoid confusion, we won't refer to LPS. Also, the sentence on Line 89 has been modified.

Line 296. Says that the polarization was only characterized for one North/South position at the center. Figure 1. Is the test setup used to get measurements over the in-orbit range of scan mirror viewing angles? Table 2 has 1 for SMA position. What about the East / West characterization? Was the assembly moved to vary the mirror scan angles?

→ No, the GEMS polarization test on the ground has been done for a very limited environment. This is why it is difficult for us to understand the nature of the variation in polarization for the north-south or east-west directions. Although not shown here, the change in PF with regards to the scan mirror angle in the East-West direction, which BATC provided as a model-based, was very small. However, the model results are not necessarily the actual values.

Line 238 "The polarization error caused by changes in total ozone is less than those caused by other changes." Figure 4. Could the authors comment on why Figure 4 does not have any (or maybe very small) dependence on TOZ? I would expect that the ozone would selectively shield the shorter channels with higher ozone absorption from the surface and clouds and thus produce wavelength-dependent changes similar in magnitude to the albedo and surface pressure changes as ozone amounts

increase. That is, alter the relative amounts of single scattered, multiple scattered and reflected radiances.

→ Fig. 4 shows the effect of each parameter on polarization error change with both the polarization state of the atmosphere and instrument considered. This is not to say that ozone does not affect polarization (the change in DOLP with and without consideration of ozone in radiative transfer models is large and significant compared to other trace gases). Since polarization is primarily affected by scattering, the change in polarization error with ozone accounted for was relatively small compared to other variables such as geometry. This suggests that we may be able to reduce the dimensionality of the LUT (e.g. using fixed total ozone amount) in the future to improve the effectiveness of the calculation.

Figure 4 and Figure 8. Figure 4 shows errors versus SZA and SVA of 1% or more. Figure 8 does not show corrections larger than 0.1%. (Are the units in Figure 4, 8, 11 and 12 all in % error in radiances?) Was the range of cases used to construct Figure 8 much less varied than the real cases in Figures 11 and 12? Particularly in SZAs?

→ Yes, the figures mean polarization error [%]. The synthetic data presented in Fig. 8 is for January 15, 2016 at 03 UT. The IOT period presented in Figs. 11, 12 is July 25, which has a large difference in the position of the sun (The figure below shows the change of SZA over a day, corresponding to the IOT period data shown in Fig. 11). From Fig. 4, it can be seen that when SZA and VZA are 30 degrees, the smaller the RAA, the larger the polarization error.

[Figure]

SZA distribution as presented in Fig. 11.

---

## Author Comment (AC2)

**Response to Reviewer #3**

We appreciate your very meaningful comments.

It gave us a deeper understanding of what we overlooked and didn't take into account, which enriched the manuscript.

General comments:

Including a discussion on the potential for validation the polarization correction with GEMS observational data would strengthen the discussion. For instance, would it be possible to compare the L1 data with and without corrections to other reference satellites? (Perhaps solar/view geometries could be chosen where the polarization effect is largest). Or could derived L2 products based on the corrected and uncorrected L1 could be compared with ground measurements.

→ Yes, verification of polarization correction is very difficult. We tried to find ways to utilize other reference satellites, as you suggested. We have tried to benchmark our method against GSICS method (of course, we don't think our criteria and method are perfect as GSICS), matching the geometry as much as possible and selecting cloud-free areas to compare with TROPOMI, etc. However, since polarization correction is not the only factor that determines the radiometric accuracy of GEMS, we could not conclude that the difference in radiance spectrum with the reference satellites is due to polarization correction. However, as you mentioned, it is necessary to continue to evaluate GEMS observation using other reference satellites.

Given the limitations in the pre-launch characterization, such as only measuring the center position, and other complexities, were L2-based correction methods considered?

→ A correction based on L2 has not yet been considered. As I mentioned at the end of the paper, I believe that algorithm testing of L2 with and without polarization correction would allow us to track spatial variation in polarization error (especially with products that can be strongly influenced by polarization spectrum feature).

Please clarify what was demonstrated with the actual GEMS data (see related specific comments below)

→ In my opinion, the information we were able to obtain from the actual GEMS data confirms the quantitative variation in polarization error that might have been taken for granted, and suggests that the increased polarization error at dawn and evening may affect each product. Indeed, the impact of this variation in polarization on L2 needs to be analyzed in future studies, which we plan to do.

Please include more details on the reference frame transformation methodology (see below for detailed comments)

→ Revised. we described it more detail with additional figure for each coordinate in payload.

Specific comments:

35: "high peak of curvature of polarization error": Do you mean in the spectral region with a sharp spectral feature in the polarization sensitivity? Please clarify.

→ Revised. We have corrected the sentence to make it a bit clearer that the peaks and curvature in the GEMS polarization coefficient cover the high wavelength region.

38: "diurnal variation for the spatial distribution of polarization error confirmed."   My understanding is that the diurnal variation is based on the calculated Stokes combined with the pre-launch measurement parameters, not the GEMS observations.

→ As you know, the polarization error is determined by the combination of the polarization state of the atmosphere and instrument. The DOLP of incident light on the GEMS get varied depends on the geometry at each observation, resulting in a spatiotemporal variation of the polarization error.

58: Could include the spectral sampling here (although I realize it is included in Table 1).

→ Revised. We have added "0.2 nm spectral sampling" to the GEMS spectral information description .

62: I think along with accuracy, stability should be emphasized as well, since, with a polarization-sensitive instrument, the diurnal signal can change as the solar/view angles change throughout the day.

→ Revised. As you suggest, we agree that the accuracy and stability of the observations are important factors in the output. We've added "stable" to the sentence.

93: This statement makes it sound like VLIDORT is the only RTM that can perform RTM in this spectral range. Perhaps more justification can be offered by citing some associated validation work for the model.

→ Revised. We cited references that did comparative verification work with VLIDORT (Escribano et al., 2019; Korkin et al., 2020). Further, we additionally cited the paper on VLIDORT's simulation uncertainty in the last discussion section (Castellanos et al., 2018).

131: Aside from a lower signal from off-nadir positions, could the polarization sensitivity vary as well depending on scan mirror angle?

→ Yes, it can also change according to the change of SMA. (For example, PF and PA can be slightly different for the E/W direction by changing the SMA. Although such a test was not performed in the polarization test of GEMS, according to the model result according to the angle change provided by BATC, the change according to E/W direction was very small even N/S direction were slight high; See below figure). But, the exact result is not known because it was only the result of the model and not the actual measurement result.

[Figure]

Fig. The variability of LPS(=PF) ratio with regards to E/W and N/S direction. The results are from BATC model. It didn't include in manuscript, but you can check it out for reference.

133: "ideal state" is a bit confusing. I recommend removing this sentence.

→ Revised. Yes, the expression "ideal" seems a bit confusing because it is a result measured under limited conditions. We removed that sentence.

160: Please provide more description on reference frame transformation methodology. For instance, it is not clear to me what the difference is between the instrument and boresight reference frame. Some suggestions:

A figure showing the geometry including definitions of the various reference frames. A working example (maybe as an appendix if the authors feel this breaks up the flow too much)

→ Revised. In the supplement, we have attached an image of the coordinate system that is transformed to define the reference frame in each part of the payload, even though the coordinate axis that defined the local meridian plane is not the same as the coordinate axis that defined the local meridian plane. (However, the details have been reinterpreted and simplified due to the confidential issues).

200: "with assumed cloud albedo to be 0.8" to "with an assumed cloud albedo of 0.8"

→ Revised.

Figure 3: spectral range is slightly different than the GEMS spectral range. Perhaps make them consistent or explain the discrepancy.

I think you meant Table 3. We simulated the RTM for LUTs with an additional 10 nm at each end of the GEMS spectral range. For consistency and to avoid confusion, we have noted it to 300-500, which corresponds to the GEMS spectral range as you suggested.

244: These statements are a bit confusing to me. In the previous section, the sensitivity of TOA radiance to ozone was shown to be smaller than the other parameters considered. Here, the authors state that polarization sensitivity plays a crucial role for ozone. Can you clarify?

→ Fig. 4 shows the effect of each parameter on polarization error change with both the polarization state of the atmosphere and instrument considered. This is not to say that ozone does not affect polarization (the change in DOLP with and without consideration of ozone in radiative transfer models is large and significant compared to other trace gases). Since polarization is primarily affected by scattering, the change in polarization error with ozone accounted for was relatively small compared to other variables such as geometry. This suggests that we may be able to reduce the dimensionality of the LUT (e.g. using fixed total ozone amount) in the future to improve the effectiveness of the calculation.

[Figure]

Fig. The changes in the DOLP simulated for the molecular atmosphere when each of O3, SO2, NO2, HCHO, and O2-O2 is individually neglected.

276: Perhaps change "ideal" to "complete" or "comprehensive"

→ Revised

322: Figure S1 and S2 seem to be missing.

→ Figure S1 and S2 are presented in 'Supplement file'. Since another figure was added, it is moved to Fig. S3 and S4.

Fig. 6: The degree of linear polarization used seems low. Was the solar geometry limited to around noon? Please add more details about the times of day/solar angle ranges, since this would greatly impact these values.

→ Yes, the synthetic data was simulated for the January 15, 2016 at 03:00 UTC. It is the noon time zone at Seoul. We have also specified the time information in Figure 6.

323: What RTM input parameters were used to simulate the clouds?

→ Clouds were simulated assuming a Mixed Lambertian cloud rather than a Mie cloud. OMI LER was applied for surface reflectivity, and GEOS-CHEM simulations were used for trace gases (O3, NO2, SO2, HCHO).

330: "exhibit sharp curvature in PF." Can you explain the significance of your choice of wavelengths. Do you expect larger errors due to the instrument's spectral sampling over these features?

→ As shown in Fig 9, the sharply increasing polarization error near certain wavelengths (349.6, 432.0, 454.6, and 494.8 nm) is reflected in the shape of the overall observed radiance spectrum, which can affect the results of retrieval using those wavelengths. In particular, errors in the radiance spectrum can be more significant when using the specific wavelength bins presented rather than utilizing a broader range of spectra. We believe that the sampling interval (0.2 nm of GEMS) allows polarization errors due to variations in these wavelength-dependent polarization errors to be more finely reflected in the observed spectrum.

354: What are "dump points"?

→ We agree that "dump point" is not a common phrase. We changed the wording to "raggedness points".

358: It would be interesting to understand the impact of the pre-launch measurement uncertainty on the polarization correction. Perhaps the authors could mention that this was not considered or that it will be considered in the future if that is the case.

→ Prelaunch measurement uncertainty has not been considered, so this may be a future consideration. The paragraph was unnecessarily descriptive and somewhat confusing, so we have added to future considerations in the Discussion.

Fig. 11, 365: Perhaps it would be helpful to clarify that you are plotting the ratio in Eq 1 as a percent difference (if that is the case).

→ Revised, we clarified that polarization error is Percent difference.

Technical corrections

358: typo: change "shist" to "shift"

→ Revised.
91: Recommend changing "comprised" to "included"

→ Revised.

---

## Author Comment (AC3)

We appreciate your very meaningful comments.
It gave us a deeper understanding of what we overlooked and didn't take into account, which enriched the manuscript.

Section 1, paragraph 3
The authors introduce two methods to deal with polarized light. One requires a scrambler and the other requires a PMD. The discussion of the two seems asymmetric because the authors state GEMS does not have a PMD but there is no mention of the scrambler. The authors could mention that a scrambler is difficult to implement in large aperture instruments such as GEMS. There is at least a third method for dealing with atmospheric polarization, and perhaps more. sensor optics can be designed in such a way that they are relatively insensitive to the polarization state of the incoming radiation. The design can be aided by including a polarization compensator in the optical train. The purpose of the compensator is to offset the polarization sensitivty caused by the remaining optical train in the sensor. It is likely that such an approach was not practical or effective for the GEMS viewing conditions, but the authors should acknowledge that there are more than two approaches to reducing polarization sensitivity.

→ Thank you. We were only focusing on the two well-known methods. By presenting a third method, we were able to enrich the content. Also, as you suggested, we made an additional comment about the difficulty of implementation on large apertures like GEMS.

Section 1, paragraph 4
It may be clearer to say, "In terms of a similar approach the MODIS and VIIRS instruments also lack both scramblers and PMDs. Polarization characteristics are measured during pre-launch testing ..."

→ Revised, we have changed in manuscript as you suggested.

Section 1, paragraph 5
The first sentence is somewhat awkward. The authors say, "we describe a newly developed polarization correction algorithm" after summarizing the GEMS polarization correction approach in the previous paragraph. The wording of this first sentence implies that there is yet another, new, approach that is different than the one summarized in paragraph 3. It would be better to simply say, "we describe the polarization correction algorithm for GEMS."
In the second sentence the language is again unclear. Are the authors trying to say that the GEMS algorithm is unique for considering clear-sky conditions, or that older approaches only considered clear-sky conditions and the GEMS algorithm now considers partly clouded conditions? Please use a few extra words so there is no misinterpretation of what you are trying to say.

→ Revised. In order to avoid confusing expression, we revised it to be concise as you suggested. The second sentence was meant that the polarization correction algorithm takes into account the polarization error in the cloud region as well as the clear sky region by including the cloud retrieval algorithm. We have reorganized the sentence.

Section 2.2, line 126
Please define 'SMA'.   Is it Scan Mirror Assembly or Scan Mirror Angle? The definition should be stated explicitly the first time the abbreviation is used.

→ Revised, it denotes "Scan Mirror Assembly (SMA)". I wrote down the full word of the SMA acronym.

Section 2.2, line 130
The sentence beginning "However, since the signal ..." is difficult to understand.   I recommend defining central position in a separate sentence to make this sentence easier to understand.

→ Revised, we have separated complex sentence expressions into two sentences.

Section 2.2
The authors treat the results of Figure 2 as the true polarization characteristic for GEMS. Yet these results may include the characteristics of the polarizer response as a function of wavelength.   I do not expect the authors to know the details of the analysis performed by Ball, but they should acknowledge that the imperfect polarization of the source may not be fully accounted for in these results. It is not clear if the authors are saying the mirror coating and stray light represent features in the PF spectrum that have been captured or errors in that spectrum.   The authors should clarify this.   This is also a useful point in the paper to discuss why these features are so important.   It is in fact the spectral dependence of the PF rather than the absolute level that affects the final science products.   If the PF was a flat line at 1% or 2% the authors may not be considering a polarization correction at all. Are the retrieval algorithms not immune to wavelength-indpendent polarization sensitivity?

→ Revised. We clarified that we mean the spectrum of PF/PA polarization error with mirror coating and stray light. We further clarified that due to their non-uniform polarization characteristics, they respond with non-uniform intensity depending on the wavelength, which can lead to the performance of the retrieval algorithm.

Section 3.1, line 156
It would help the reader if they could see a figure showing the meridian plane and the instrument plane together.   It is hard to visualize these angles and their coordinate frames based simply on the verbal description.

→ Revised. In the supplement, we have attached an image of the coordinate system that is transformed to define the reference frame in each part of the payload, even though the coordinate axis that defined the local meridian plane is not the same as the coordinate axis that defined the local meridian plane. (However, the details have been reinterpreted and simplified due to the confidential issues).

Section 3.1, final paragraph
This paragraph raises some questions about the GEMS polarization correction.   If the LUTs require as independent parameters surface pressure, albedo, and trace gas concentrations, how are these derived at Level 1B since this product generally does not have such information?   Some of

these parameters are discussed in the next section, but it is worth noting in this paragraph that the polarization correction uses preliminary estimates for these parameters rather than final retrieved quantities.

A single sentence is devoted to derivation of the surface pressure. This deserves as much or more discussion than the derivation of ozone or terrain height, and I recommend expanding the LER discussion in Section 3.3.2 to include characterization of the reflecting surface.

→ Surface pressure, albedo, and trace gas concentration, which are LUT configuration parameters, use the auxiliary climatological data. As mentioned in Section 3.3, total ozone amount is adopted from climatological data based on OMI data, surface pressure is conversed from the terrain height of ETOPO-2, and the cloud area is estimated through a separate retrieval algorithm. Surface reflectivity is from GOME-2 LER database. The process of deriving the surface pressure is estimated by converting it into atmospheric pressure by the barometric formula using Terrain Height data (ETOPO-2).

Section 3.3.1, paragraph 2
It is useful to cite the DOI for the OMI data. In this way you need not discuss the details of exactly which product you used (for instance, that it is from Collection 3). The DOI can be obtained from the GES DISC.

→ Revised. We cite the DOI for OMI data (http://doi.org/10.5067/Aura/OMI/DATA204) as founded from GES DISC.

Section 3.3.3, line 310
Simply say that the correction algorithm is not very sensitive to the reflecting pressure under cloud-free conditions and that use of a terrain height pressure results in a negligible error.

→ Revised. We additionally mentioned in the manuscript as you suggested.

Section 4.1
The second largest source of error in the polarization correction (after the GEMS characterization) is probably the VLIDORT simulation of TOA radiances. The authors provide no estimation of this error. Instead they assume the dominant error comes from simplifications in the model assumptions (used to generate the LUT). The LUT errors may be significant, but that does not mean the full VLIDORT simulation errors are negligible. Clearly, the results shown in the right hand side of Figure 7 represent an underestimation of these errors. I view this as a major flaw in this paper. Can the authors provide an independent estimation of the missing error? How much larger could the true errors be? There is a brief mention of aerosol errors toward the end of Section 4.1, and there is a limited discussion of model deficiencies in Section 5, paragraph 2. This is the kind of discussion I am talking about, and it needs to be expanded.

→ Revised, I agree that it is important to present the factors that cause polarization errors. The uncertainty caused by the simplification and interpolation of the LUT were quantified and presented. In addition, we presented the uncertainty by citing a paper that calculated the simulation error by benchmarking VLIDORT. (>0.1%)(Castellanos et al., 2018).the polarization errors caused by the use of LUTs did not affect significantly in defining DOLP and polarization angles. Perhaps, we think that

the large factor of the residual error is the estimation error for the cloud surface pressure to input value to the LUT and the fact that aerosols are not considered. As you said, we have improved and refined the Discussion Section by adding parts that can be quantitatively presented about the factors that can cause these errors. This is same as follows comment for Section 5.2 below.

Figure 8 caption
It will be clearer to label these as polarization error rather than radiance difference. Using the term radiance difference leaves the reader asking: difference between what and what?

→ Revised, we have corrected expression as polarization error to avoid confusing the reader. We also corrected the labeling of the x-axis in the figure.

Figure 9 caption
The left vs. right description in the figure caption is reversed.

→ Revised

Section 4.1, last sentence
I am not familiar with the term "dump point." Please choose different terminology or explain what is meant by this phrase.

→ Revised, we totally agree that "dump point" is not a common phrase. We changed the wording to "raggedness points".

Section 4.2, first paragraph
Please consider rewriting or removing this paragraph. It lacks a point and seems out of place. The first problem is that it is not obvious to the average reader why SNR and shift/squeeze (not shist/squeeze) should be a consideration for the polarization effect. You do eventually explain the problem of wavelength registration, but in a complicated way. Please use simple statements such as, "The polarization and polarization corrections affect the spectral structure of radiances. Since the wavelength registration of each Earth scene relies on radiance spectral structure these results can be affected." Also, please explain the effect on SNR. The second issue with this paragraph is that the authors raise the problems but do not resolve them. How large are the errors in wavelength registration? How much is SNR affected? Is there anything that can be done to reduce the uncertainty? If the authors simply wish to say that these are problems to be considered then this discussion best belongs in Section 5 as 'future work'.

→ Revised. We agree with your suggestions. Overall, SNR is independent of polarization calibration. (We did mention that SNR error remains after correction because it is independent of polarization correction. Therefore, we removed the misleading SNR statement). The next main point is, as you said, the issue of wavelength calibration. As the polarization error (PF/PA) is a function of wavelength, it is necessary to allocate the wavelength value at the exact location. The polarization correction algorithm assumes that the wavelength calibration was done well in the previous L1B step, and the error due to wavelength calibration is difficult to calculate here. These points have been moved to Section 5 and simply mentioned, rather than being discussed in this section.

Figure 12 caption
To be clear, these are polarization errors prior to correction.

→ Revised. We clearly mentioned that the polarization error illustrated in the box-plot is the polarization error corrected by the polarization algorithm.

Section 4.2, second paragraph

This is an important discussion about how the polarization error can alias into diurnal variation in the data products. It starts off well, but ends up in the wrong place. The authors' conclusion is that it is important to apply the polarization correction properly. That is stating the obvious. It will be a more useful discussion if the authors can estimate the residual diurnal errors (after correction). For example, Figure 12 could instead be a plot of the radiometric uncertainty as a function of time-of-day. This requires the authors to have some idea of the uncertainties in the combined ground characterization + VLIDORT correction. As I have stated previously, the lack of such uncertainty is a major omission in this paper.

→ Yes, it would be obvious, but what I wanted to show with the actual GEMS data was that there was an unwanted polarization error that varied in diurnal variation, and wanted to quantify it. Due to our limited information, we do not know how much polarization error is actually contained in every pixel in the GEMS domain (we do not know the residual error after polarization correction). What we show in this section is the amount of polarization error that can be corrected with the information we have, and at least the polarization error included in the radiation amount can be corrected as shown. As for the uncertainty in the polarization error, as you suggested, we will quantify it more in Section 5 and discuss the possibilities further.

→ Yes, we missed the important point in this section at the end. We consider it important to suggest the possibility that diurnal variability in non-constant polarization error may affect the performance of L2 products. This is why it is important to do polarization correction, and it is necessary to analyze its effect in future L2 products.

Section 5, paragraph 2

The discussion here of areas of uncertainty in the polarization correction is very important. These uncertainties deserve to be quantified rather than simply listed. The reader cannot assess the validity of the polarization correction without some estimation of these uncertainties. I understand it is a lot of work to come up with reasonable uncertainties, but that does not mean the authors can ignore the issue. The lack of PF information as a function of SMA angle is a good example. The authors offer a plausible approach to improving the characterization, and it is understandable they do not include such improvements in this paper. But they should include estimates of the uncertainties as a result of the SMA spatial dependence. Even though the authors do not have access to the original test data, there are ways of estimating the degree of S and P polarization from a reflecting aluminum surface given the incidence and view angles. It's true that we do not know exactly how the scan mirror is coated, but the goal here is to bound the error rather than to improve the characterization.

→ Revised, As answered for Section 4.1 above, we discussed uncertainty a bit more in the Discussion section. If aerosol is not taken into account and clear skies are assumed, the actual atmospheric polarization state can be misunderstood as large, and then overcorrection. Along the same line, If the cloud surface pressure is estimated lower/higher than the actual value, the polarization effect can be over/under corrected. In addition, the factor of shift/squeeze of the GEMS wavelength is also mentioned here (since the polarization error of GEMS is a function of wavelength, it is important to apply the correction at the correct wavelength position). Based on the results of the BATC model (as below figure), the change in the polarization spectrum of 350 to 400 nm can increase up to 6 times (in the N/S direction). Assuming the model results, a maximum polarization error can be 0.4% (worst case). Then, the polarization error of ~0.3% can be remained even after correction with the polarization correction algorithm proposed in this study.

[Figure]

Fig. The variability of LPS(=PF) ratio with regards to E/W and N/S direction. The results are from BATC model. It didn't include in manuscript, but you can check it out for reference.

Section 5, paragraph 3
I cannot understand what the authors are saying here. Please rewrite this paragraph and simplify the sentences.

→ Revised. The wording of the sentence was a little unclear. Since the polarization characteristics are wavelength-dependent, it was said that the calculation accuracy could be improved by evaluating the effect of the presence or absence of polarization correction of the L2 product with respect to the spatiotemporal polarization correction effect. The sentence has been rewritten by simplifying it so that it is not complicated.

---

## Author Comment (AC4)

We appreciate your very meaningful comments.

It gave us a deeper understanding of what we overlooked and didn't take into account, which enriched the manuscript.

General comments:

1. This paper discusses both atmospheric and instrumental polarizations, but some sentences are confusing. Please clarify throughout the paper.

   → Revised, we've clarified to avoid confusion.

2. This study used the pre-flight instrumental polarization sensitivity on the central point of the GEMS instrument. However, it is important to note that the polarization sensitivity can vary with different angles and over time due to different solar and viewing zenith angles. Although estimating the changes in the polarization sensitivity is challenging, it is important to assess whether the algorithm effectively corrects polarization effects on the radiance in real. If it is not easy to do so, it is recommended to provide quantitative values by showing the improvement of GEMS L1B and L2 products after polarization corrections or doing a sensitivity test. Additionally, the limitations of the algorithm should be discussed more when the algorithm is applied to real GEMS data.

   → Yes, that's right. The main purpose of this study is to define the polarization sensitivity of the GEMS in the prelaunch test (although limited information can be used), and to apply the polarization correction algorithm in near real time adopt that atmospheric polarization vary according to the observation geometry. Although the minimum quantitative value that can be polarization corrected was presented, there are still remained uncertainty sources. Therefore, the possibilities that could cause these errors were discussed together. I totally agree about the importance of looking at the impact on L2 product. However, it is hard to run L2 algorithms in the breadth of this study. We plan to analyze it with other L2 developers in the next phase of our future study.

3. The polarization correction algorithm utilizes spectra calculated by one of the radiative transfer models (RTMs), VLIDORT. Since a model is not perfect, it is important for the authors to ensure that VLIDORT simulates Stokes parameters well by providing references. To address this, the author should describe VLIDORT in an independent chapter of Section 3 or somewhere.

   → Revised, We cite references where VLIDORT has been benchmarked and studies that have tested its simulation performance against the GOME-2 PMD, and further describe VLIDORT in Section 3.

4. It is necessary to ensure that the sentences and terms are clear and unambiguous to avoid any confusion for the readers.

   → Revised, we've modified some ambiguous wording to improve clarity.

Specific comments:

Line 29: Is it the polarization axis, not angle?

→ Revised. Polarization axis is right expression.

47: Please provide specific a wavelength region to describe the polarization effects.

→ Revised. Polarization effects within UV-Vis regions.

48: Typo Mischenko -> Mishchenko. Please check all references.

→ Revised.

52: There are TROPOMI and OMPS nadir mappers with the same objectives.

→ Revised, we mentioned OMPS and TROPOMI, and added references.

62-78: Please clarify whether instrumental or atmospheric polarization is referred to here and throughout the paper. For example, the PMD is a device to measure instrumental polarization? However, the authors explained that GEMS does not have a PMD to measure atmospheric polarization states.

→ Revised. PMD measures the fractional polarization of the atmosphere. Wee clarified in the sentence.

126: What does SMA stand for? Regarding one of the general comments, polarization effects with different SMA angles should be discussed.

→ Revised, it denotes "Scan Mirror Assembly (SMA)". I wrote down the full word of the SMA acronym.

128: Please describe the physical meaning of PA like PF.

→ Revised, we explain the meaning of PF and PA.

150-154: Chi is the polarization axis, but chi_LMP is the polarization angle? Also, please explain how the polarization angle (chi_LMP) is calculated by Eq (2).

→ Sorry for the confusion. The description of definitions of $\phi$ and $\chi$ were reversed. $\phi$ is polarization axis, and $\chi$ is polarization angle via IRP.

175-185: Define all notations and their meanings. The quaternion matrix and multiplication might be unfamiliar to many readers. It would be helpful to provide an overall explanation of the quaternion matrix and multiplication in the Appendix.

→ Revised. As you pointed out, quaternions may be unfamiliar to others. We've added a more detailed explanation of what quaternions mean and what was lacking in that paragraph.

189: Figure 3 just shows the overall flow of processes from original GEMS L1B to corrected GEMS L1B, but it does not present the algorithm flow chart. It would be more helpful to show how parameters in Eq (1) are derived using input data in Figure 3. In addition, please provide the meanings of box shapes in Figure 3.

→ As you suggested, I thought deeply about showing how the parameters are derived using each variable, but I think it would be too complicated and confusing to implement it in the flow chart, such as each variable is directly entered into Eq. (1). We would like to keep the current structure. please, confirm. Instead, we modified the "polarization correction algorithm" that was stated in the small square to "polarization correction equation", as it could have been misleading since the whole flow chart means the flow of polarization correction algorithm. And We added the corresponding variable in Atmosphere and Instrument, which are colored red and blue. Perhaps the description of the box shapes you suggested was referring to the red and square boxes, rather than a description of what the shape of each part does in the process, so we added a caption explaining what each box means that related polarization parameters for atmosphere and instrument.

192: In the re-process, the authors use GEMS L2 products for polarization correction, but satellite products have large uncertainty. It is worth considering if the method of using GEMS L2 products can achieve the same performance compared to polarization correction using climatological data.

→ Yes, you are right. We also have a process of reprocessing using L2 output derived from the same pixel, rather than climate data. Although it was mentioned in the main text that this content is not dealt with in depth, as you said, the accuracy and uncertainty of the L2 output should be acquired. In particular, the performance of the L2 cloud will be the most influential during the reprocessing process. In particular, the performance of the L2 cloud will be the most influential during the reprocessing process. We tested using L2 Total Ozone, which has relatively stable and high performance. As already suggested in the text, the change in total ozone amount has little effect on polarization, and the distribution does not change significantly compared to climatology. So, the difference was very small between NRT and reprocess (almost same). Since L2 algorithms are still being improved by developers, it will be possible to improve the reprocessing process through continuous comparison/evaluation in the future.

202: How well does a radiative transfer model including VLIDORT simulate atmospheric polarization effects? Regarding the general comment, it would be helpful to provide a basic description of the model with references.

→ A previous study (Choi et al., 2020) showed good agreement between simulation by VLIDORT and GOME-2 PMD (especially for clear sky).

218: The light could be polarized by liquid water and water vapor, which are abundant in climate and weather conditions. Why are they not considered?

→ In the polarization correction algorithm, the polarization of the cloud area is also considered. While we haven't simulated mie clouds or ice particles, we do have a process for not incorrectly correcting cloud areas by assuming a clear sky. When tested in previous studies, the difference in the change in DOLP between Mie and Lambertian clouds was not significant. The figure below shows an example of a comparison between Mie and Lambertian clouds, along with the cloud height (COD), from Choi et al., 2020.

[Figure]

Fig. Average changes in the DOLP across the spectrum (a) for the various CODs and (b) calculated using Mie theory (red line) and the Lambertian top surface (blue line)

229: Can the GEMS slit function be assumed as a Gaussian function? Is there any error resulting from a different shape of the slit function?

→ The SRF of GEMS is close to the Gaussian function. The variation of SRF according to the change of wavelength is also very small. Therefore, the applied Gaussian function (FWHM 0.6nm) is similar to the actual SRF, and the error due to the shape of the slit function is small. Although, it is necessary to reconstruct by applying the actual GEMS SRF in the future.

233: Please define what I_obs and I_true represent.

→ Revised.

283: GOME-2 already has a coarse spatial pixel size of 80 km x 40 km, and it is difficult to achieve a finer resolution than its own spatial resolution. Therefore, despite interpolation to a finer spatial resolution, it does not make the surface LER more accurate.

→ Yes, you are right, as you said, the spatial resolution of GOME-2 is sparse, and even if interpolated to match the resolution of GEMS, it cannot accurately represent the reflectance of each GEMS pixel. In this study, we use climatological GOME-2 LER database as input, but if a GEMS-optimized surface reflectance database is created in the future, it would be good to use it. Thanks for your comment.

322 The normalized radiance is not shown anywhere in this paper, and Figure S2 seems to only show Q and U components without cloud. The authors should clarify a sentence and provide supporting figures.

→ We added the radiance spatial distribution map to Supplements Figure 5. Since we have added a figure for Radiance (I), we have changed in the manuscript from "normalized radiance" to "radiance".

327: It is difficult to distinguish clouds in the figures. Therefore, it is hard to understand cloud effects on polarization. It would be helpful to show clouds in figures.

→ Along the same line to the answer above, we have added an illustration of reflectivity in Supplementary Figure 5. We have defined the region with a reflectivity of 0.2 or less as the clear sky region, which is mentioned in the caption.

Figure 10: The difference is mainly related to interpolation? Rather than interpolation, it seems to be other reasons because the difference is too large.

→ Yes. The most dominant reason for the difference is the interpolation of the LUTs as shown in Fig. 10: the gaps look like a big, but it's actually a 4-digit decimal order difference. As we discuss in the Discussion, linear interpolation causes these small differences and is one of the sources for the residual polarization error.

372: Figure 5 just showed climatological ozone, not diurnal variation.

→ Sorry for confusing. Not Fig. 5, but Fig. 4. Corrected accordingly.

Table 1: Please correct a unit of the spatial resolution (not km).

→ Revised. We modified it to $km^2$.

---

## Referee Report (RR1)

This work discusses a polarization correction scheme for GEMS to reduce the impact of the instrument's polarization sensitivity on the observed radiance by illustrating its impacts on synthetic data. The revised manuscript clarifies many of the concepts and provides a more thorough discussion on future work towards maturing this capability. Please address the few remaining points of clarifications and minor technical errors:

- Line 33: – Please change to "...radiance spectrum can include a polarization error of 2%"
- 70: "fractional polarization of atmosphere" isn't a generally well-known quantity. It may be worth defining in terms of the Stokes parameters (½(1-Q/I))
- 83: Remove "may"
- 96: Change "of LUT" to "in the LUT"
- 136: Please clarify. I think you are trying to say that the measurements were done at the nadir position because the signal was lower at off-nadir positions. Is that correct? Depending on your confidence in the models, perhaps the authors would consider mentioning the PS angular variability predictions to give a sense of the order of magnitude of these variations (without including the plots).
- 150: This description is a little unclear. Perhaps something like "...radiance response is non-uniform across wavelength due to the non-uniform PF spectrum, which can lead to degraded performance." (I believe you are missing a word, possibly "degraded", here.)
- 160: The details that were added to this section are helpful. However, could you clarify why so many transforms are needed to get from the instrument GEMS reference frame to the GEMS boresight?
- 234: "and enable" to "enabling"
- 385: Perhaps "raggedness points" is not a common descriptive term either. I recommend "spectral features" instead.

---

## Author Response (AR2)

**Response to Reviewer #2**

We appreciate your very meaningful comments. Here we have prepared a response and explanation to your suggestion, and have incorporated it into the main text.

Section 2, Line 131

The authors are inadequately describing the SMA and its role in GEMS observations. Readers are left with the impression that GEMS polarization characteristics are only known for one view angle, which is not the case. Please explain that deviations in SMA from zero cause the entire view to shift north and south and are not part of nominal operations.

→ Revised. Yes, I agree with the opinion that this can easily lead to misunderstanding. As you suggested, we mentioned in the manuscript a deviation of the SMA from 0˚ position induces a shift in the entire view toward the north or south, thereby diverging from nominal operations. As you know, unlike TEMPO where polarization testing was performed for 5 SMA positions (each different location), GEMS only performed polarization testing for SMA 0 degrees. This is why it is difficult for us to accurately determine polarization characteristics in the north/south directions.

Line 138

Please clarify what you mean by decreasing signals with increasing distance. What distance is increasing?

→ Revised. This implies that the response sensitivity to the polarization source from the integrating sphere decreases not only in the North-South direction but also across the wavelength spectrum on the CCD, making it difficult to reliably detect a consistent signal. We mentioned this in more detail in the manuscript.

Section 3, Line 174

Please comment on the size of these rotations. Is the polarizer angle mostly aligned to the meridian plane or are the differences signficant? Most readers don't want to wade through the minutia of your coordinate transformations to figure that out.

→ Revised. In the boresight frame, the polarizer angle of 0 degrees is aligned with the eastern direction of GEMS observations. As a result, the transformation from LMP to IRP is similar to a counterclockwise rotation of approximately 90 degrees (Fig. 6). we mentioned these in the manuscript.

Section 3, Line 212

I suggest the authors start a new subsection here. As written, the text jumps without a break from a discussion of the coordinate transformation of the polarization angles into a description of data processing.

→ Revised. As you suggested, the paragraph below shows the configuration flow of the overall algorithm, so we started session 3.2 with the configuration of the polarization correction algorithm.

Section 3, Line 221

Please describe how you handle partially clouded scenes. Do you use an independent pixel approximation and mix clear-sky reflectivity at the terrain and 0.8 albedo clouds at a lower pressure? Or do you place all clouds at the terrain height? I see no other description of cloud reflectivity in Section 3. This should be described in a bit more detail somewhere.

→ Revised. We described it in more detail. Each individual observed pixel is considered like partial cloud using the IPA method, which assumes that photon movement occurs only vertically, not horizontally, and that it consists of cloud-free surface impact and partial clouds. The cloud reflectivity was assumed to be a typical 0.8.

Section 4, Lines 374-384

The discussion here is important. In my opinion, it is more important than the preceding discussion of LUT lookup errors, which are of little interest to readers because they can be minimized by simply making the LUT denser. Of more interest are the modeling errors, and that discussion belongs in this section. The authors allude to errors associated with Lambertian cloud assumptions and aerosol loading, but then defer to Section 5. Discussion of model uncertainties is central to the evaluation of algorithm performance and belongs here.

In my opinion this paper would would be more useful to readers if the authors repeated some of the investigations of errors cited in Choi et al using real GEMS scenes. What range of polarization errors can we expect from a range of AOD and particle height distrubutions? Likewise, how might non-Lambertian clouds change the GEMS polarization sensitivity? I think that such investigations fall within the scope of this paper and should not be deferred to future work.

→ Yes, we fully agree with your comment. LUT nodes are important, but may not be significant (As you said, it can be solved by making the spacing denser or finding the optimal node). We have additionally described in the text information about the polarization changes by aerosol height, AOD, and cloud treatment. The degree of polarization attenuation varies depending on the AOD and aerosol height. This suggests that even if aerosol influence is inherent in the cloud processing process, polarization error may be overcorrected if corrected for the clear sky without considering aerosols. The difference of effects on polarization between Lambertian cloud and Mie cloud (non-Lambertian clouds) is relatively low (slightly higher by Mie clouds for very high cloud).

Section 4.2, Line 393 and Figure 11 caption.

I think the authors are simply discussing the polarization correction. The terms "degree of corrected polarization error" and "corrected polarization error" are confusing. Please change this or, if I am mistaken, explain exactly the quantity you are describing and plotting.

→ Sorry for the confusion. We use the expression "corrected polarization error" consistently.

Section 4.2

The discussion of diurnal variation in the polarization error seems to belong in Section 3. After all, these are not residual errors. The authors are claiming these errors are corrected by their algorithm.

Section 4.2 is merely describing the geophysical variation caused by viewing conditions, similar to the discussion in Section 3.2.

→ Yes, what is shown here using actual GEMS data is to present and show the daily change in polarization error that can occur due to observation conditions, as you mentioned. This is the polarization error that can be corrected with the proposed polarization correction algorithm. It is unknown how much residual error remains. As you mentioned, it refers to changes depending on observation conditions, but considering that it can be applied and presented to actual data, we think it would be better to cover it in 4.2.

We appreciate your very meaningful comments. Here we have prepared a response and explanation to your suggestion, and have incorporated it into the main text.

Line 33: – Please change to "...radiance spectrum can include a polarization error of 2%"

→ Revised. We modified the sentence as you suggested.

Line 70: "fractional polarization of atmosphere" isn't a generally well-known quantity. It may be worth defining in terms of the Stokes parameters (1⁄2(1-Q/I))

→ Revised. We rephrase as "Stokes fraction (Q/I)"

Line 96: Change "of LUT" to "in the LUT"

→ Revised.

Line 136: Please clarify. I think you are trying to say that the measurements were done at the nadir position because the signal was lower at off-nadir positions. Is that correct? Depending on your confidence in the models, perhaps the authors would consider mentioning the PS angular variability predictions to give a sense of the order of magnitude of these variations (without including the plots)

→ As other reviewer also has made similar points, a deviation of the SMA from 0° position induces a shift in the entire view toward the north or south, thereby diverging from nominal operations. This implies that the response sensitivity to the polarization source from the integrating sphere decreases not only in the North/South direction but also across the wavelength spectrum on the CCD, making it difficult to reliably detect a consistent signal. We explained that part in a little more detail in the text.

Line 150: This description is a little unclear. Perhaps something like "...radiance response is non-uniform across wavelength due to the non-uniform PF spectrum, which can lead to degraded performance." (I believe you are missing a word, possibly "degraded", here.)

→ Revised.

Line 160: The details that were added to this section are helpful. However, could you clarify why so many transforms are needed to get from the instrument GEMS reference frame to the GEMS boresight?

→ Incident light on the payload sequentially passes through each part frame inside various instruments on its path to reach the CCD, and each part frame has its own axis. Therefore, the axis is converted step by step until it is converted to the final GEMS boresight frame.

Line 234: "and enable" to "enabling"

→ Revised.

Line 385: Perhaps "raggedness points" is not a common descriptive term either. I recommend

"spectral features" instead.

→ Revised.